# ♣ CLOVER ♣: Probabilistic Forecasting with Coherent Learning Objective Reparameterization

**Kin G. Olivares**[*]                                                *kigutie@amazon.com*
*Amazon Forecasting Science, New York*

**Geoffrey Négiar**[*†]                                             *geoff@theforecastingcompany.com*
*The Forecasting Company*

**Ruijun Ma**[*]                                                    *ruijunma@amazon.com*
*Amazon Forecasting Science, New York*

**O. Nganba Meetei**                                               *nganba@gmail.com*
*Peloton, New York*

**Mengfei Cao**                                                    *mfcao@amazon.com*
*Amazon Forecasting Science, New York*

**Michael W. Mahoney**                                            *zmahmich@amazon.com*
*Amazon Forecasting Science, New York*

**Reviewed on OpenReview:** *https://openreview.net/forum?id=q7YXEbFOAt*

## Abstract

Obtaining accurate probabilistic forecasts is an operational challenge in many applications, such as energy management, climate forecasting, supply chain planning, and resource allocation. Many of these applications present a natural hierarchical structure over the forecasted quantities; and forecasting systems that adhere to this hierarchical structure are said to be coherent. Furthermore, operational planning benefits from the accuracy at all levels of the aggregation hierarchy. However, building accurate and coherent forecasting systems is challenging: classic multivariate time series tools and neural network methods are still being adapted for this purpose. In this paper, we augment an MQForecaster neural network architecture with a modified multivariate Gaussian factor model that achieves coherence by construction. The factor model samples can be differentiated with respect to the model parameters, allowing optimization on arbitrary differentiable learning objectives that align with the forecasting system's goals, including quantile loss and the scaled Continuous Ranked Probability Score (CRPS). We call our method the *Coherent Learning Objective Reparametrization Neural Network* (CLOVER). In comparison to state-of-the-art coherent forecasting methods, CLOVER achieves significant improvements in scaled CRPS forecast accuracy, with average gains of 15%, as measured on six publicly-available datasets.

---

[*] Equal contribution.
[†] Work completed at Amazon Forecasting Science, where Geoffrey interned as a UC Berkeley Ph.D. candidate.

| Method | Coherent End-to-End | Multivariate Series Inputs | Multivariate Output Distribution | Arbitrary Learning Objective |
|---|:---:|:---:|:---:|:---:|
| `PERMBU` (Ben Taieb et al., 2017) | ✗ | ✗ | ✗ | ✗ |
| `Bootstrap` (Panagiotelis et al., 2023) | ✗ | ✗ | ✗ | ✗ |
| `Normality` (Wickramasuriya, 2023) | ✗ | ✗ | ✓ | ✗ |
| `DPMN` (Olivares et al., 2023) | ✓ | ✗ | ✓ | ✗ |
| `HierE2E` (Rangapuram et al., 2021) | ✓ | ✓ | ✗ | ✗ |
| `CLOVER` (ours) | ✓ | ✓ | ✓ | ✓ |

Table 1: Coherent forecast methods' desirable properties.

# 1 Introduction

Obtaining accurate forecasts is an important step for planning in complex and uncertain environments, with applications ranging from energy to supply chain management (Chen et al., 2022; Wolff et al., 2024), from transportation to climate prediction (Hong et al., 2014; Gneiting & Katzfuss, 2014; Makridakis et al., 2022). Going beyond point forecasts such as means and medians, probabilistic forecasting provides a key tool for predicting uncertain future events. This involves, for example, forecasting that there is a 90% chance of rain on a certain day or that there is a 99% chance that people will want to buy fewer than 100 items in a certain store in a given week. Providing more detailed predictions of this form permits finer uncertainty quantification. This, in turn, allows planners to prepare for different scenarios and allocate resources according to their anticipated likelihood and cost structure. This can lead to better resource allocation, improved decision-making, and less waste.

In many forecasting applications, there exist natural hierarchies over the quantities one wants to predict, such as energy consumption at various temporal granularities (from monthly to weekly), different geographic levels (from building-level to city-level to state-level), or retail demand for specific items (in a hierarchical product taxonomy). Typically, most or all levels of the hierarchy are important: the bottom levels are key for operational short-term planning, while higher levels of aggregation provide insight into longer-term or broader trends. Moreover, probabilistic forecasts are often desired to be coherent (or consistent) to ensure efficient decision-making at all levels (Hong et al., 2014; Jeon et al., 2019). Coherence is achieved when the forecast distribution assigns zero probability to forecasts that do not satisfy the hierarchy's constraints (Ben Taieb et al., 2017; Panagiotelis et al., 2023; Olivares et al., 2023) (see Definition 2.1). Designing an accurate model, capable of leveraging information from all hierarchical levels while enforcing coherence is a well-known and challenging task (Hyndman et al., 2011).

The hierarchical forecasting literature has been dominated by two-stage reconciliation approaches, where univariate methods are first fitted and later reconciled towards coherence. For many years, most research has focused on mean reconciliation (Hyndman et al., 2011; 2024; Vitullo, 2011; Hyndman et al., 2016; Dangerfield & Morris, 1992; Wickramasuriya et al., 2019; Mishchenko et al., 2019). More recent statistical methods consider coherent probabilistic forecasts through variants of the bootstrap reconciliation technique (Ben Taieb et al., 2017; Panagiotelis et al., 2023) or the clever use of the properties of the Gaussian forecast distributions (Wickramasuriya, 2023). A detailed hierarchical forecast review is provided by Athanasopoulos et al. (2024). Large-scale applications of hierarchical forecasting require practitioners to simplify the two-stage reconciliation process by favoring *end-to-end* approaches that simultaneously fit all levels of the hierarchy, while still achieving coherence. The end-to-end approach refers to training a model constrained to achieve coherence by directly optimizing for accuracy. End-to-end methods offer advantages such as reduced complexity, improved computational efficiency, and improved adaptability by streamlining the entire forecasting pipeline into a single, unified model. More importantly, end-to-end models generally achieve better accuracy compared to two-stage models that are first trained independently for optimized accuracy and then made coherent through various reconciliation approaches (Rangapuram et al., 2021; Olivares et al., 2023).

To our knowledge, only three methods produce coherent probabilistic forecasts and allow models to be trained end-to-end: Rangapuram et al. (2021), Olivares et al. (2023), and Das et al. (2023). (There is also parallel research on hierarchical forecasts with relaxed constraints (Han et al., 2021; Paria et al., 2021; Kamarthi et al., 2022; 2024; Umagami et al., 2023), but we limit our analysis of this line of work since our focus is on strictly coherent forecasting methods). In particular, Olivares et al. (2023) considers a finite mixture of Poisson distributions that captures correlations implicitly through latent variables and does not leverage cross-time series information. Donti et al. (2021) achieves coherence by constructing probabilistic forecasts for aggregated time series through the completion of equality. On the other hand, Rangapuram et al. (2021) leverages multivariate time series information but achieves coherence through a differentiable projection layer, which could degrade forecast accuracy: it does not directly model correlations between the multivariate outputs, but rather couples them through its projection layer. Dedicated effort is still necessary to capture these hierarchical relationships to improve forecast accuracy. Such probabilistic methods can benefit from the ability to optimize for arbitrary loss functions by differentiating samples, as demonstrated by Rangapuram et al. (2021). The capacity to optimize any loss computed from the forecast samples can help align the forecasting system's goals with the neural network's learning objective.

Summarized, an ideal hierarchical forecasting method should satisfy several desiderata: 1) be coherent end-to-end; 2) model a joint multivariate probability distribution, capturing the intricate relationships between series within the hierarchy; 3) leverage cross-time series information and accurately reflect these relationships; and 4) generate differentiable samples, to enable the method to optimize for arbitrary learning objectives that align with the forecasting system's goals. In this paper, we present a method that satisfies all these ideal properties; see Table 1 for a summary.

We introduce the *Coherent Learning Objective Reparameterization Neural Network* (`CLOVER`), a method for producing probabilistic coherent forecasts that satisfies all the desired properties stated above. Our **main contributions** are the following.

1. We design a multivariate Gaussian factor model that achieves forecast coherence exactly by construction, representing the joint forecast distribution over the bottom-level series. This factor model is generic and can augment most neural forecasting models with minimal modifications, with the condition that all the hierarchy's series are available during training and inference.

2. In contrast to other joint distribution models, our factor model samples' differentiability—enabled by the reparameterization trick—offers greater versatility, allowing optimization of a wider range of learning objectives beyond negative log likelihood. We leverage this versatility by training the model using both the Continuous Ranked Probability Score (CRPS) and the Energy Score (Matheson & Winkler, 1976). For hierarchical forecasting tasks, aligning the training objective with the CRPS evaluation metric leads to significant improvements in accuracy.

3. We combine the multivariate factor model with an augmented `MQCNN` architecture (Wen et al., 2017; Olivares et al., 2022c) into the *Coherent Learning Objective Reparameterization Neural Network* (`CLOVER`). Our augmented architecture accepts vector autoregressive (VAR) inputs through a multi layer perceptron module.

4. `CLOVER` achieves state-of-the-art performance in six publicly available benchmark datasets, demonstrating substantial improvement in both probabilistic and mean forecast accuracy. On average, `CLOVER` outperforms the second-best method by 15 percent in probabilistic forecasting accuracy and delivers a 30 percent increase in mean forecast accuracy.

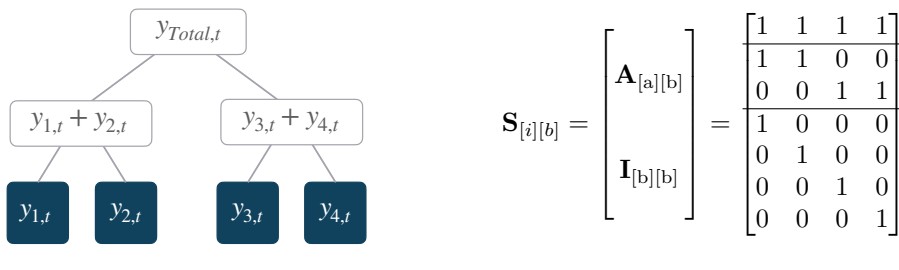

(a) Graph representation

(b) Matrix representation

Figure 1: A simple time series hierarchical structure with $N_a = 3$ aggregates over $N_b = 4$ bottom time series. Figure 1a shows the disaggregated bottom variables with blue background. Figure 1b (right) shows the corresponding hierarchical aggregation constraints matrix with horizontal lines to separate levels of the hierarchy. We decompose our evaluation throughout the levels.

## 2 Hierarchical Forecast Task.

**Notation.** We introduce the hierarchical forecast task following Olivares et al. (2023). We denote a hierarchical multivariate time series vector by $\mathbf{y}_{[i],t} = \left[ \mathbf{y}_{[a],t}^\top \mid \mathbf{y}_{[b],t}^\top \right] \in \mathbb{R}^{N_a + N_b}$, where $[i] = [a] \cup [b]$, $[a]$, and $[b]$ denote the set of full, aggregate, and bottom indices of the time series, respectively. There are $\|[i]\| = N_a + N_b$ time series in total, with $\|[a]\| = N_a$ aggregates from the $\|[b]\| = N_b$ bottom time series, at the finest level of granularity. We use $t$ as a time index. In our notation, we keep track of the shape of tensors using square brackets in subscripts. Since each aggregated time series is a linear transformation of the multivariate bottom series, we write the hierarchical aggregation constraint as

$$\mathbf{y}_{[i],t} = \mathbf{S}_{[i][b]} \mathbf{y}_{[b],t} \iff \begin{bmatrix} \mathbf{y}_{[a],t} \\ \mathbf{y}_{[b],t} \end{bmatrix} = \begin{bmatrix} \mathbf{A}_{[a][b]} \\ \mathbf{I}_{[b][b]} \end{bmatrix} \mathbf{y}_{[b],t}. \tag{1}$$

The aggregation matrix $\mathbf{A}_{[a][b]}$ represents the collection of linear transformations for deriving the aggregates and sums the bottom series to the aggregate levels. The hierarchical aggregation constraints matrix $\mathbf{S}_{[i][b]}$ is obtained by stacking $\mathbf{A}_{[a][b]}$ and the $N_b \times N_b$ identity matrix $\mathbf{I}_{[b][b]}$.

For a simple example, consider $N_b = 4$ bottom-series, so $[b] = \{1, 2, 3, 4\}$ and $y_{Total,t} = \sum_{i=1}^4 y_{i,t}$. Figure 1 shows an example of such hierarchical structure, where the multivariate hierarchical time series is defined as:

$$\mathbf{y}_{[a],t} = [y_{Total,t}, \ y_{1,t} + y_{2,t} \ y_{3,t} + y_{4,t}]^\top, \qquad \mathbf{y}_{[b],t} = [y_{1,t}, \ y_{2,t}, \ y_{3,t}, \ y_{4,t}]^\top. \tag{2}$$

**Probabilistic Forecast Task.** Consider historical temporal features $\mathbf{x}_{[b][:t]}^{(h)}$, known future information $\mathbf{x}_{[b][t+1:t+N_h]}^{(f)}$, and static data $\mathbf{x}_{[b]}^{(s)}$, forecast creation date $t$ and forecast horizons in $[t+1 : t+N_h]$. A classic forecasting task aims to estimate the following conditional probability [1]:

$$\mathbb{P}_{[i]} \left( \mathbf{Y}_{[i],t+\eta} \mid \mathbf{x}_{[b][:t]}^{(h)}, \ \mathbf{x}_{[b][t+1:t+N_h]}^{(f)}, \ \mathbf{x}_{[b]}^{(s)} \right) \qquad \text{for } \eta = 1, \cdots, N_h. \tag{3}$$

The hierarchical forecasting task augments the forecast probability in Eqn. 3 with coherence constraints in Eqn. 2.1 (Ben Taieb et al., 2020; Panagiotelis et al., 2023; Olivares et al., 2022c), by restricting the probabilistic forecast space to assign zero probability to non-coherent forecasts. Definition 2.1 formalizes the intuition, stating that the distribution of a given aggregate random variable is exactly the distribution defined as the aggregates of the bottom-series distributions through the summation matrix $\mathbf{S}_{[i][b]}$.

---

[1]The conditional independence assumption across $t$ in Eqn. 3, maintains the computational tractability of the forecasting task, and is an assumption used by most forecasting methods.

**Definition 2.1.** Let $(\Omega_{[b]}, \mathcal{F}_{[b]}, \mathbb{P}_{[b]})$ be a probabilistic space (on the bottom-level series $\mathbf{Y}_{[b]}$), with sample space $\Omega_{[b]}$, event space $\mathcal{F}_{[b]}$, and $\mathbb{P}_{[b]}$ a forecast probability. Let $\mathbf{S}_{[i][b]}(\cdot) : \Omega_{[b]} \mapsto \Omega_{[i]}$ be the linear transformation implied by the aggregation constraints matrix. A **coherent forecast** space $(\Omega_{[i]}, \mathcal{F}_{[i]}, \mathbb{P}_{[i]})$ satisfies

$$\mathbb{P}_{[i]}\left(\mathbf{S}_{[i][b]}(\mathcal{B})\right) = \mathbb{P}_{[b]}(\mathcal{B}), \tag{4}$$

for any set $\mathcal{B} \in \mathcal{F}_{[b]}$ and its image $\mathbf{S}_{[i][b]}(\mathcal{B}) \in \mathcal{F}_{[i]}$.

**Hierarchical Forecast Scoring Rule.**    In this work and most of the hierarchical forecast literature, the performance of probabilistic forecasts is primarily evaluated by the Continuous Ranked Probability Score (CRPS), e.g. Ben Taieb et al. (2017); Rangapuram et al. (2021); Olivares et al. (2023); Panagiotelis et al. (2023); Das et al. (2023); Wickramasuriya (2023). The CRPS between a target $y$ and distributional forecast $Y$ is defined as

$$\mathrm{CRPS}\left(y, Y\right) = \mathbb{E}_Y\left[\,|Y - y|\,\right] - \frac{1}{2}\mathbb{E}_{Y,Y'}\left[\,|Y - Y'|\,\right], \tag{5}$$

where $Y'$ is distributed as $Y$, but is independent of it (Matheson & Winkler, 1976; Gneiting & Raftery, 2007).

CRPS is commonly used because it is a strictly proper scoring rule (Gneiting & Raftery, 2007) and is agnostic to model and distributional assumptions. The metric also gives a summary of performance on all quantile forecasts (Laio & Tamea, 2007), as

$$\mathrm{CRPS}(y, Y) = 2 \int \mathrm{QL}_q(y, F_Y^{-1}(q))dq, \tag{6}$$

where $F_Y^{-1}(q)$ is the $q$-quantile of variable $Y$. The $q$-quantile loss is defined as

$$\mathrm{QL}_q(y, F_Y^{-1}(q)) = q(y - F_Y^{-1}(q))_+ + (1 - q)(F_Y^{-1}(q) - y)_+. \tag{7}$$

## 3  Methodology

In this section, we describe our main method, the *Coherent Learning Objective Reparameterization Neural Network* (`CLOVER`). It consists of a multivariate probabilistic model, an underlying neural network structure, and a coherent end-to-end model estimation procedure.

### 3.1  Coherent Probabilistic Model

Our predicted probabilistic forecasts at all hierarchical levels are jointly represented by a Gaussian factor model. Our neural network maps the known information (past, static and known future) to the location, scale, and shared factor parameters, and the forecasted factor model parameters are designed to model correlations between the bottom-level series, while conditioning on all known information. Our factor model[2] combined with the coherent aggregation in Eqn. 10 directly estimates the multivariate probability of bottom-level series $\mathbf{y}_{[b][t+1:t+N_h]}$ conditioning on historical, known-future, and static covariates $\mathbf{x}_{[b][:t]}^{(h)}$, $\mathbf{x}_{[b][t+1:t+N_h]}^{(f)}, \mathbf{x}_{[b]}^{(s)}$, i.e.,

$$\mathbb{P}\left(\tilde{\boldsymbol{Y}}_{[i][t+1:t+N_h]} \mid \mathbf{x}_{[b][:t]}^{(h)}, \ \mathbf{x}_{[b][t+1:t+N_h]}^{(f)}, \mathbf{x}_{[b]}^{(s)}\right) = \mathbb{P}\left(\mathbf{S}_{[i][b]}\hat{\boldsymbol{Y}}_{[b][t+1:t+N_h]} \mid \hat{\boldsymbol{\mu}}_{[b][h],t}, \ \hat{\boldsymbol{\sigma}}_{[b][h],t}, \hat{\mathbf{F}}_{[b][k][h],t}\right). \tag{8}$$

At a given forecast creation date $t$, the model uses the location $\hat{\boldsymbol{\mu}}_{[b][h],t} \in \mathbb{R}^{N_b \times N_h}$, scale $\hat{\boldsymbol{\sigma}}_{[b][h],t} \in \mathbb{R}^{N_b \times N_h}$ and shared factor $\hat{\mathbf{F}}_{[b][k][h],t} \in \mathbb{R}^{N_b \times N_k \times N_h}$ parameters, along with samples from standard normal variables $\mathbf{z}_{[b],\eta} \sim \mathcal{N}(\mathbf{0}_{[b]}, \mathbf{I}_{[b][b]})$, and $\boldsymbol{\epsilon}_{[k],\eta} \sim \mathcal{N}(\mathbf{0}_{[k]}, \mathbf{I}_{[k][k]})$ to compose the following multivariate variables for each horizon:

$$\hat{\mathbf{y}}_{[b],\eta,t} = \hat{\boldsymbol{\mu}}_{[b],\eta,t} + \mathrm{Diag}(\hat{\boldsymbol{\sigma}}_{[b],\eta,t})\mathbf{z}_{[b],\eta,t} + \hat{\mathbf{F}}_{[b][k],\eta,t}\boldsymbol{\epsilon}_{[k],\eta,t}, \qquad \eta = 1, \cdots, N_h. \tag{9}$$

---

[2]Early work on factor forecast models augmenting neural networks done by Wang et al. (2019) does not ensure coherence.

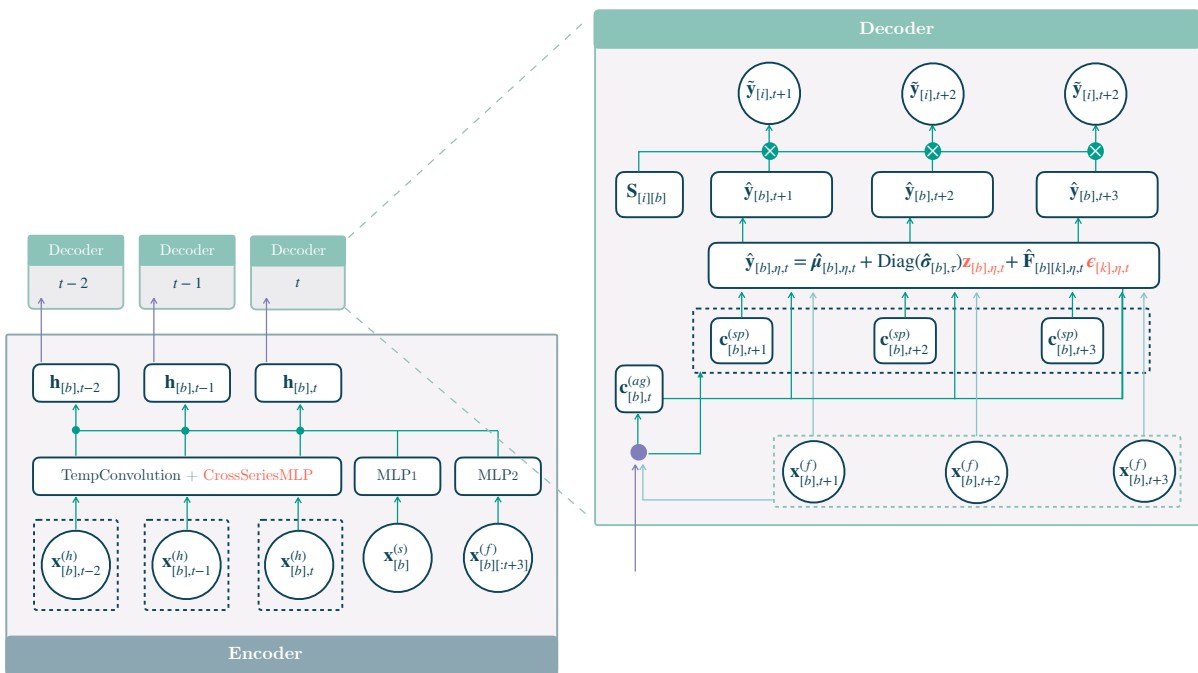

Figure 2: The *Coherent Learning Objective Reparameterization Neural Network* is a Sequence-to-Sequence with Context network that uses dilated temporal convolutions as the primary encoder and multilayer perceptron based decoders for the creation of the multi-step forecast. `CLOVER` coherently aggregates the samples of the factor model $\tilde{\mathbf{y}}_{[i],\eta,t} = \mathbf{S}_{[i][b]}\hat{\mathbf{y}}_{[b],\eta,t}$. We mark in red the standard normal samples that are parameter-free, the reparameterization trick allows to apply backpropagation through the factor model outputs. `CLOVER` extends upon the univariate `MQCNN`, through the cross series multi layer perceptron.

After sampling from the multivariate factor, we *coherently aggregate* the clipped outputs of the network,

$$\tilde{\mathbf{y}}_{[i],\eta,\tau} = \mathbf{S}_{[i][b]} \left( \hat{\mathbf{y}}_{[b],\eta,\tau} \right)_{+}. \tag{10}$$

The shared factors enable the factor model to capture the relationships across the disaggregated series, and the covariance structure of the disaggregated series follows:

$$\mathrm{Cov}\left(\hat{\mathbf{y}}_{[b],\eta,t}\right) = \mathrm{Diag}(\hat{\boldsymbol{\sigma}}^2_{[b],\eta,t}) + \hat{\mathbf{F}}_{[b][k],\eta,t}\hat{\mathbf{F}}^{\top}_{[b][k],\eta,t}. \tag{11}$$

**Definition 3.1.** Any bottom-level multivariate distribution can be transformed into a coherent distribution through **coherent aggregation** [3]. Given a sample $\hat{\mathbf{y}}_{[b]} \sim \mathbb{P}_{[b]}$, a coherent $\mathbb{P}_{[i]}$ distribution can be constructed with the following sample transformation

$$\tilde{\mathbf{y}}_{[i]} = \mathbf{S}_{[i][b]} \left( \hat{\mathbf{y}}_{[b]} \right). \tag{12}$$

In other words, it is enough to aggregate the bottom-level forecasts in a bottom-up manner. We include in Appendix A a proof of the approach's coherence property and the covariance structure.

## 3.2  Neural Network Architecture

As mentioned in Section 1, the factor model can augment most neural forecasting models if all series in the hierarchy are available during training and inference, as this is a sufficient condition to obtain the factor model parameters. Appendix F shows an augmented subset of NeuralForecast models (Olivares et al., 2022b),

---

[3]Coherent aggregation can be thought of a special case of bootstrap reconciliation (Panagiotelis et al., 2023) that only relies on a bottom-level forecast distribution.

while the main paper focuses on the `MQCNN`-based model (Wen et al., 2017; Olivares et al., 2022c). We select `MQCNN` for its outstanding performance in multi-step forecasting problems. Our architecture based on `MQCNN` has a main encoder which consists of a stack of dilated temporal convolutions, and it is applied to historical information for all series. In addition, it uses a global multi layer perceptron (MLP) to encode the static and future information. The encoder at time $t$ in Eqn. 13, is applied to each disaggregated series:

$$\mathbf{h}_{[i],t}^{(h)} = \text{TempConvolution}\left(\left[\mathbf{S}_{[i][b]}\mathbf{x}_{[b][:t]}^{(h)}\right]\right)^{(4)}$$

$$\mathbf{h}_{[b]}^{(s)} = \text{MLP}_1\left(\mathbf{x}_{[b]}^{(s)}\right) \tag{13}$$

$$\mathbf{h}_{[b],t}^{(f)} = \text{MLP}_2\left(\mathbf{x}_{[b][t+1:t+N_h]}^{(f)}\right).$$

We use a residual cross series MLP to capture vector autoregressive relationships in the hierarchy with minimal modifications to the architecture [5]:

$$\mathbf{h}_{[b],t}^{(h)} = \text{CrossSeriesMLP}\left(\mathbf{h}_{[i],t}^{(h)}\right). \tag{14}$$

`CLOVER` uses a two-stage MLP decoder: the first decoder summarizes information in the horizon-agnostic context $\mathbf{c}_{[b],t}^{(ag)}$ and the horizon-specific context $\mathbf{c}_{[b][h],t}^{(sp)}$; and the second stage decoder transforms the contexts into the factor model parameters $\left(\hat{\boldsymbol{\mu}}_{[b][h],t}, \hat{\boldsymbol{\sigma}}_{[b][h],t}, \hat{\mathbf{F}}_{[b][k][h],t}\right)$. Eqn. 15 describes the operations:

$$\mathbf{h}_{[b],t} = \left[\mathbf{h}_{[b],t}^{(h)}, \mathbf{h}_{[b]}^{(s)}, \mathbf{h}_{[b],t}^{(f)}\right]$$

$$\mathbf{c}_{[b],t}^{(ag)} = \text{MLP}_3\left(\mathbf{h}_{[b],t}\right) \qquad \text{and} \qquad \mathbf{c}_{[b][h],t}^{(sp)} = \text{MLP}_4\left(\mathbf{h}_{[b],t}\right) \tag{15}$$

$$\left(\hat{\boldsymbol{\mu}}_{[b][h],t}, \hat{\boldsymbol{\sigma}}_{[b][h],t}, \hat{\mathbf{F}}_{[b][k][h],t}\right) = \text{MLP}_5\left(\left[\mathbf{c}_{[b][h],t}^{(sp)}, \mathbf{c}_{[b],t}^{(ag)}, \mathbf{x}_{[b][t+1:t+N_h]}^{(f)}\right]\right).$$

For the last step, the network composes the factor model samples using Eqn. 9 and aggregates them in Eqn. 10.

### 3.3   Learning Objective Reparameterization

Let $\boldsymbol{\theta}$ be a model that resides in the class of models $\Theta$ defined by the model architecture. Here, $\theta$ can be thought as a mapping from the model feature space to the parameter set for all target horizons, we have

$$(\hat{\boldsymbol{\mu}}_{[b][h],t}, \hat{\boldsymbol{\sigma}}_{[b][h],t}, \hat{\mathbf{F}}_{[b][k][h],t}) = \boldsymbol{\theta}(\mathbf{x}_{[b][:t]}^{(h)}, \mathbf{x}_{[b][t+1:t+N_h]}^{(f)}, \mathbf{x}_{[b]}^{(s)}). \tag{16}$$

Let $\hat{Y}_{i,\eta,t}(\boldsymbol{\theta})$ be the random variable parameterized by $\boldsymbol{\theta}$. In some problems, multi-step coherent forecasts for multiple items are needed (e.g., in retail business, coherent regional demand forecasts are required for each product). Let $u$ be the index of such an item within an index set $\{1, \cdots, N_u\}$ of interest, and let $\tilde{Y}_{u,i,\eta,t}(\boldsymbol{\theta})$ be the coherent forecast random variable for the target $y_{u,i,t+\eta}$. As we explain in Appendix I, using the reparameterization trick (Kingma & Welling, 2013), within the class of parameters $\Theta$ defined by the neural network architecture, we optimize for either CRPS

$$\min_{\boldsymbol{\theta} \in \Theta} \sum_{u,i,\eta,t} \text{CRPS}\left(y_{u,i,t+\eta}, \tilde{Y}_{u,i,\eta,t}(\boldsymbol{\theta})\right) = \sum_{u,i,\eta,t} \mathbb{E}\left[|y_{u,i,t+\eta}, \tilde{Y}_{u,i,\eta,t}|\right] - \frac{1}{2}\mathbb{E}\left[|\tilde{Y}_{u,i,\eta,t} - \tilde{Y}_{u,i,\eta,t}'|\right]. \tag{17}$$

or the Energy Score

$$\min_{\boldsymbol{\theta} \in \Theta} \sum_{u,t} \text{ES}\left(\mathbb{P}_{u,t,[i][h]}, \tilde{Y}_{u,t,[i][h]}(\boldsymbol{\theta})\right) = \sum_{u,t} \mathbb{E}\left[||y_{u,t,[i][h]} - \tilde{Y}_{u,t,[i][h]}||_2^\beta\right] - \frac{1}{2}\mathbb{E}\left[||\tilde{Y}_{u,t,[i][h]} - \tilde{Y}_{u,t,[i][h]}'||_2^\beta\right]. \tag{18}$$

We train the model using stochastic gradient descent (`Adam` (Kingma & Ba, 2014)) with early stopping (Yao et al., 2007). Appendix C contains details of the network's optimization and hyperparameter selection.

---

[4]Temporal exogenous data only aggregates the target signal, other features (e.g., calendar) are maintained without aggregation.

[5]Approach also inspired by regional forecast at Amazon's Supply Chain Optimization Technologies (Savorgnan et al., 2024).

### 3.4 Discussion

Here, we discuss differences between `CLOVER` (our method) with two coherent end-to-end probabilistic forecasting baselines in `HierE2E` (Rangapuram et al., 2021) and `DPMN` (Olivares et al., 2023).

The `HierE2E` method of Rangapuram et al. (2021) is "too general." It consists of an augmented `DeepVAR` neural network model (Flunkert et al., 2017) that produces marginal forecasts for all hierarchical series. `HierE2E` claims to be more general than hierarchical forecasting, since it is designed to enforce any convex constraint satisfied by the forecasts; due to the constraining operation in the method, it has to revise the optimized forecasts. It does not leverage the specifics of the hierarchical constraints, which are more structured than a general convex constraint. `HierE2E` produces samples from independent Gaussian distributions for each time-series in the hierarchy; since the samples are not guaranteed to be hierarchically coherent, `HierE2E` couples samples by projecting them onto the space of coherent probabilistic forecasts. Both the sampling operation (Kingma & Welling, 2013) and the projection are differentiable, allowing the method to be trained end-to-end; `HierE2E` restricts to optimize likelihood or CRPS. Both `HierE2E` and `CLOVER` can allow different distribution choices, since Gaussians can be replaced by any distribution that can be sampled in a differentiable way, i.e., almost any continuous distribution (Ruiz et al., 2016; Figurnov et al., 2018; Jankowiak & Obermeyer, 2018). In Rangapuram et al. (2021), the projection operator ensures coherence, and correlations between bottom-levels are learned only by optimizing the neural network. In contrast, `CLOVER` produces forecasts for bottom-level series only, while relying on common factors to encode correlations. This removes the need to forecast at all levels simultaneously, therefore, reducing computational requirements if we are only interested in a subset of the aggregates.

On the other hand, the `DPMN` baseline (Olivares et al., 2023) is "too restrictive," in particular, as a Poisson mixture can be prone to distribution misspecification problems. It is known that when a probability model is misspecified, optimizing log likelihood is equivalent to minimizing Kullback–Leibler (KL) divergence with respect to the true probabilistic distribution, KL divergence measures change in probability space, while optimizing CRPS is equivalent to minimizing the Cramer-von Mises criterion (Gneiting & Raftery, 2007), which quantifies the distance with respect to the probability model in the sample space. `CLOVER`'s learning objective for the probabilistic model is resilient to distributional misspecification (Bellemare et al., 2017). Moreover, `CLOVER` can be optimized to accommodate other evaluation metrics of interest.

Finally, `DPMN` estimates the covariance among time series, but does not take advantage of the multivariate input when encoding historical time series. Similarly to other `ARIMA`-based baselines, on specific hierarchical benchmark datasets such as `Traffic`, `DPMN` produces suboptimal bottom-series forecasts. We improve the encoder for historical time series by adding a CrossSeriesMLP after the Temporal convolution encoder, which bridges the accuracy gap between `HierE2E` and our `MQCNN`-based approach.

## 4 Empirical Evaluation

In this section, we present our main empirical results. First, we describe the empirical setup. Second, we evaluate `CLOVER` and compare it with state-of-the-art hierarchical forecast models. Third, we present the results of the ablation study that further analyze the source of improvements in variants of `CLOVER`.

### 4.1 Setting

**Datasets.** We analyze six qualitatively different public datasets: `Labour`, `Traffic`, `Tourism-S`, `Tourism-L`, `Wiki`, and `Favorita`, each requiring significant modeling flexibility due to their varied properties. The `Favorita` dataset is the largest dataset evaluated which includes count and real-valued regional sales data, with over 340,000 series. The `Tourism-S` and `Tourism-L` datasets, report quarterly and monthly visitor numbers to Australian regions, respectively, and they are grouped by region and travel purpose. The `Traffic` dataset contains daily highway occupancy rates from the San Francisco Bay Area, featuring highly correlated series with strong Granger causalities. The smallest `Labour` dataset tracks monthly Australian employment by status, gender, and geography, the series in this dataset are highly cointegrated. Lastly, the `Wiki` dataset summarizes daily views of online articles by country, topic, and access type. We provide more dataset details in Appendix D.

| Dataset | # Items ($N_u$) | Bottom ($N_b$) | Levels | Aggregated ($N_a + N_b$) | Time range | Frequency | Horizon ($N_h$) |
|---|---|---|---|---|---|---|---|
| Labour | 1 | 32 | 4 | 57 | 2/1978-12/2019 | Monthly | 12 |
| Traffic | 1 | 200 | 4 | 207 | 1/2008-3/2009 | Daily | 1 |
| Tourism-S | 1 | 56 | 4 | 89 | 1998-2006 | Quarterly | 4 |
| Tourism-L | 1 | 304 | 4/5 | 555 | 1998-2016 | Monthly | 12 |
| Wiki | 1 | 150 | 5 | 199 | 1/2016-12/2016 | Daily | 7 |
| Favorita | 4036 | 54 | 4 | 93 | 1/2013 - 8/2017 | Daily | 34 |

Table 2: Summary of publicly-available data used in our empirical evaluation.

**Evaluation metrics.** Our main evaluation metric is the mean scaled CRPS from Eqn. 5 defined as the score described in Eqn. 19, divided by the sum of all target values. Let $\boldsymbol{l}^{(g)}$ be a vector of length $N_a + N_b$ consisting of binary indicators for a hierarchical level $g$, where for each $j \in [i]$, $l_j^{(g)} = 1$ if aggregated series $j$ is included in the hierarchical level $g$, and 0 otherwise. Then sCRPS for the hierarchical level $g$ is defined as

$$\text{sCRPS}\left(\mathbf{y}_{[i][t+1:t+N_h]}, \ \tilde{Y}_{[i][h],t} \mid \boldsymbol{l}^{(g)}\right) = \frac{\sum_{i=1}^{N_a+N_b} \left(\sum_{\eta=1}^{N_h} \text{CRPS}(y_{i,t+\eta}, \tilde{Y}_{i,\eta,t})\right) \cdot l_i^{(g)}}{\sum_{i=1}^{N_a+N_b} ||\mathbf{y}_{i,[t+1:t+N_h]}||_1 \cdot l_i^{(g)}}. \tag{19}$$

**Baseline Models.** We compare our method with the following coherent probabilistic methods: (1) DPMN-GroupBU (Olivares et al., 2023), (2) HierE2E (Rangapuram et al., 2021), (3) ARIMA-PERMBU-MinT (Ben Taieb et al., 2017), (4) ARIMA-Bootstrap-BU (Panagiotelis et al., 2023), and (5) an ARIMA. In addition, in Appendix G, we compare our method with the following coherent mean methods: (1) DPMN-GroupBU, (2) ARIMA-ERM (Ben Taieb & Koo, 2019), (3) ARIMA-MinT (Wickramasuriya et al., 2019), (4) ARIMA-BU, (5) an ARIMA, and (6) Seasonal Naive. We use the implementation of statistical methods available in the StatsForecast and HierarchicalForecast libraries (Olivares et al., 2022c; Garza et al., 2022).

## 4.2 Forecasting Results

As mentioned above, we compare the proposed model to the DPMN (Olivares et al., 2023), the HierE2E (Rangapuram et al., 2021), and two ARIMA-based reconciliation methods (Wickramasuriya et al., 2019; Panagiotelis et al., 2023). Following previous work, we report the sCRPS at all levels of the defined hierarchies; see Table 3. The reconciliation results for ARIMA are generated using HierarchicalForecast Olivares et al. (2022c), with each confidence interval calculated based on 10 independent runs. The results for HierE2E are generated based on three independent runs using hyperparameters tuned by Olivares et al. (2022c). All metrics for DPMN are quoted from Olivares et al. (2023) with identical experimental settings on all datasets.

As the *Overall* row in Table 3 shows that CLOVER improves sCRPS compared to the best alternative in five of six datasets, with gains of 27.67% on Favorita, 10.24% on Tourism-S, 4.16% on Tourism-L, 4.5% on Wiki and 54.40% in Traffic. There is a degradation of 8% in the reconciliation that performs the best on Labour, the smallest data set. For five out of six datasets, our model achieves the best or second-best sCRPS accuracy across all the levels of the hierarchy; it is important to consider that aggregate levels are much smaller in sample size for which we prefer the bottom-level measurements as an indicator of the methods' accuracy.

In Traffic, our model achieves remarkably better results than the DPMN and HierE2E baselines, which we explain by the ability to model VAR relationships accurately because of the smoothing the time series features before using them as inputs for other series. For the smallest dataset, Australian Labour, all deep learning models, including CLOVER, showed a decline in accuracy compared to statistical baselines. We hypothesize that this performance drop is due to the limited data available, as 57 monthly series may not be sufficient to effectively train complex models like deep learning architectures. Another explanation for the Labour degradation is the presence of strong trends and cycles, which can lead to artificially high correlations between the series, resulting in spurious relationships that may confuse the VAR modules.

We complement the main results with a mean forecast evaluation section in Appendix G. As we see in Table 7 the sCRPS accuracy gains are mostly mirrored in the relative squared error metric (relSE) . We qualitatively

Table 3: Empirical evaluation of probabilistic coherent forecasts. Mean *scaled CRPS* (sCRPS) averaged over 5 runs, at each aggregation level, the best result is highlighted (lower values are preferred). We report 95% confidence intervals, the methods without standard deviation have deterministic solutions.

[*] The HierE2E results differ from Rangapuram et al. (2021), here the sCRPS quantile interval space has a finer granularity of 1 percent instead of 5 percent in Rangapuram et al. (2021). We run HierE2E's best reported hyperparameters on Tourism-S with horizon=4, instead of HierE2E's original horizon=8.
[**] PERMBU-MinT on Tourism-L is unavailable because its implementation cannot naively be applied to datasets with multiple hierarchies, this is because the bottom up aggregation strategy cannot identify a unique way to obtain the upper level distributions.

| Data | Level | CLOVER (crps) | CLOVER (energy) | DPMN-GroupBU | HierE2E [*] | PERMBU-MinT [**] | Bootstrap-BU | ARIMA (not coherent) |
|---|---|---|---|---|---|---|---|---|
| Labour | Overall | 0.0074±0.0005 | 0.0075±0.0002 | 0.0102±0.0011 | 0.0171±.0003 | **0.0067±0.0001** | 0.0076±0.0001 | 0.0070 |
| | 1 (geo.) | 0.0032±0.0010 | 0.0035±0.0006 | 0.0033±0.0011 | 0.0052±.0003 | **0.0013±0.0001** | 0.0017±0.0001 | 0.0017 |
| | 2 (geo.) | 0.0054±0.0005 | 0.0054±0.0004 | 0.0088±0.0009 | 0.0181±.0003 | 0.0045±0.0001 | 0.0053±0.0001 | **0.0044** |
| | 3 (geo.) | 0.0075±0.0003 | **0.0074±0.0004** | 0.0115±0.0009 | 0.0188±.0003 | 0.0076±0.0001 | 0.0086±0.0001 | 0.0076 |
| | 4 (geo.) | 0.0137±0.0004 | 0.0138±0.0002 | 0.0173±0.0020 | 0.0262±.0004 | **0.0133±0.0001** | 0.0148±0.0001 | 0.0138 |
| Traffic | Overall | **0.0171±0.0036** | 0.0228±0.0075 | 0.0907±0.0024 | 0.0375±0.0058 | 0.0677±0.0061 | 0.0736±0.0024 | 0.0751 |
| | 1 (geo.) | **0.0026±0.0012** | 0.0064±0.0077 | 0.0397±0.0044 | 0.0183±0.0091 | 0.0331±0.0085 | 0.0468±0.0031 | 0.0376 |
| | 2 (geo.) | **0.0029±0.0014** | 0.0064±0.0067 | 0.0537±0.0024 | 0.0183±0.0081 | 0.0341±0.0081 | 0.0483±0.0030 | 0.0412 |
| | 3 (geo.) | **0.0044±0.0022** | 0.0075±0.0059 | 0.0538±0.0022 | 0.0209±0.0071 | 0.0417±0.0061 | 0.0530±0.0025 | 0.0549 |
| | 4 (geo.) | **0.0587±0.0106** | 0.0709±0.0104 | 0.2155±0.0022 | 0.0974±0.0021 | 0.1621±0.0027 | 0.1463±0.0017 | 0.1665 |
| Tourism-S | Overall | **0.0631±0.0012** | 0.0669±0.0022 | 0.0740±0.0122 | 0.0761±0.0007 | 0.0812±0.0010 | 0.0703±0.0017 | 0.0759 |
| | 1 (geo.) | **0.0268±0.0031** | 0.0304±0.0049 | 0.0329±0.0074 | 0.0400±0.0009 | 0.0375±0.0014 | 0.0335±0.0026 | 0.0354 |
| | 2 (geo.) | **0.0484±0.0019** | 0.0507±0.0036 | 0.0502±0.0058 | 0.0609±0.0012 | 0.0679±0.0016 | 0.0507±0.0023 | 0.0709 |
| | 3 (geo.) | **0.0784±0.0020** | 0.0817±0.0027 | 0.0914±0.0171 | 0.0914±0.0008 | 0.0959±0.0012 | 0.0845±0.0016 | 0.0848 |
| | 4 (geo.) | **0.0989±0.0017** | 0.1050±0.0031 | 0.1212±0.0238 | 0.1122±0.0007 | 0.1234±0.0012 | 0.1124±0.0013 | 0.1125 |
| Tourism-L | Overall | **0.1197±0.0037** | 0.1268±0.0045 | 0.1249±0.0020 | 0.1472±0.0029 | - | 0.1375±0.0013 | 0.1416 |
| | 1 (geo.) | 0.0292±0.0042 | 0.0294±0.0080 | 0.0431±0.0040 | 0.0842±0.0051 | - | 0.0622±0.0026 | **0.0263** |
| | 2 (geo.) | **0.0593±0.0049** | 0.0598±0.0059 | 0.0637±0.0032 | 0.1012±0.0029 | - | 0.0820±0.0019 | 0.0904 |
| | 3 (geo.) | **0.1044±0.0030** | 0.1077±0.0054 | 0.1084±0.0033 | 0.1317±0.0022 | - | 0.1207±0.0010 | 0.1389 |
| | 4 (geo.) | **0.1540±0.0046** | 0.1620±0.0060 | 0.1554±0.0025 | 0.1705±0.0023 | - | 0.1646±0.0007 | 0.1878 |
| | 5 (prp.) | **0.0594±0.0076** | 0.0600±0.0040 | 0.0700±0.0038 | 0.0995±0.0061 | - | 0.0788±0.0018 | 0.0770 |
| | 6 (prp.) | 0.1100±0.0049 | 0.1140±0.0034 | **0.1070±0.0023** | 0.1336±0.0042 | - | 0.1268±0.0017 | 0.1270 |
| | 7 (prp.) | **0.1824±0.0024** | 0.1934±0.0045 | 0.1887±0.0032 | 0.1955±0.0025 | - | 0.1949±0.0010 | 0.2022 |
| | 8 (prp.) | **0.2591±0.0050** | 0.2902±0.0054 | 0.2629±0.0034 | 0.2615±0.0016 | - | 0.2698±0.0004 | 0.2834 |
| Wiki | Overall | **0.2475±0.0076** | 0.3184±0.0262 | 0.3158±0.0240 | 0.2592±0.0031 | 0.4008±0.0046 | 0.2816±0.0036 | 0.3907 |
| | 1 (geo.) | **0.0823±0.0161** | 0.1536±0.0666 | 0.1709±0.0354 | 0.1007±0.0046 | 0.1886±0.0129 | 0.1630±0.0065 | 0.1981 |
| | 2 (geo.) | **0.1765±0.0154** | 0.2381±0.0171 | 0.2299±0.0241 | 0.1963±0.0037 | 0.2691±0.0073 | 0.2192±0.0043 | 0.2566 |
| | 3 (geo.) | **0.2743±0.0108** | 0.3213±0.0343 | 0.3311±0.0230 | 0.2784±0.0038 | 0.4049±0.0062 | 0.2923±0.0032 | 0.4100 |
| | 4 (geo.) | **0.2814±0.0100** | 0.3326±0.0332 | 0.3370±0.0223 | 0.2900±0.0043 | 0.4236±0.0062 | 0.2999±0.0032 | 0.4182 |
| | 5 (prp.) | **0.4231±0.0127** | 0.5466±0.1406 | 0.5098±0.0667 | 0.4307±0.0039 | 0.7177±0.0064 | 0.4339±0.0039 | 0.6708 |
| Favorita | Overall | **0.2908±0.0025** | 0.3279±0.0052 | 0.4020±0.0182 | 0.5298±0.0091 | 0.4670±0.0096 | 0.4110±0.0085 | 0.4373 |
| | 1 (geo.) | **0.1841±0.0033** | 0.2209±0.0045 | 0.2760±0.0149 | 0.4714±0.0103 | 0.2692±0.0076 | 0.2900±0.0067 | 0.3112 |
| | 2 (geo.) | **0.2754±0.0026** | 0.3112±0.0051 | 0.3865±0.0207 | 0.5182±0.0107 | 0.3824±0.0092 | 0.3877±0.0082 | 0.4183 |
| | 3 (geo.) | **0.2945±0.0025** | 0.3313±0.0053 | 0.4068±0.0206 | 0.5291±0.0129 | 0.6838±0.0108 | 0.4490±0.0098 | 0.4446 |
| | 4 (geo.) | **0.4092±0.0022** | 0.4484±0.0068 | 0.5387±0.0253 | 0.6012±0.0131 | 0.5532±0.0116 | 0.5749±0.0003 | 0.5749 |

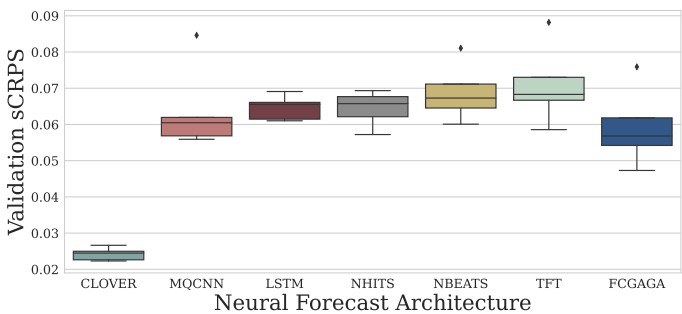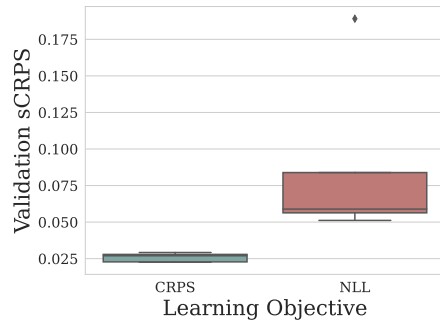

Figure 3: Ablation studies on the Bay Area `Traffic` dataset: a) In highly correlated hierarchies, VAR inputs enabled by `CLOVER` significantly improve over the univariate `MQCNN`'s accuracy. b) The factor model CRPS learning objective demonstrates clear advantages over classic negative log-likelihood. Full ablation studies described in Appendix F.

show the `CLOVER` forecast distributions for a hierarchical structure in the `Favorita` dataset in Appendix E. We also include in Appendix E a probability-probability plot comparing the similarity of the empirical and forecast distributions. As can be seen in Figure 7, the forecast distributions are qualitatively well calibrated.

### 4.3   Ablation Studies

To analyze `CLOVER`'s source of improvements, we performed two ablation studies on the `Traffic` dataset, where we investigated first the effects of the learning objective and then the effects of VAR inputs enabled by the CrossSeriesMLP. Here we report a summary and refer to the details in Appendix F.

In the first study, we explore the effects of a cross-series MLP that mimics the vector autoregressive model. We compare the `CLOVER` architecture with and without the cross-series multi layer perceptron (CrossSeriesMLP) introduced in Eqn. 14. Such a module enables the network to share information from the series in the hierarchy with minimal modifications to the architecture. Table 5 and Figure 3 show that CrossSeriesMLP improved `Traffic` forecast accuracy by 66% compared to variants without it. We attribute the effectiveness of the VAR approach to the presence of Granger-causal relationships in traffic intersections. The VAR-augmented `CLOVER` improves upon well-established univariate architectures, including `LSTM` (Sak et al., 2014), `NBEATS` (Oreshkin et al., 2020; Olivares et al., 2022a), `NHITS` (Challu et al., 2023), `TFT` (Lim et al., 2021), and `FCGAGA` (Oreshkin et al., 2021), a spatio-temporal specialized architecture.

In the second ablation study on learning objectives, we compared CRPS-based optimization, as described in Eqn. 17, with the negative log-likelihood estimation for the Gaussian factor model introduced in Section 3.1, and other likelihood-estimated distributions. Table 6 and Figure 3 show that the CRPS-optimized factor model improves forecast accuracy by nearly 60% when compared to the log-likelihood optimized model.

## 5   Conclusion

In this work, we present a novel multivariate factor forecasting model, integrated with the MQCNN neural network architecture, resulting in the *Coherent Learning Objective Reparameterization Neural Network* (`CLOVER`). Using CRPS as the learning objective, we achieve significant improvements in forecast accuracy over traditional negative log-likelihood objectives.

Our experiments on six benchmark datasets—Favorita grocery demand, Australian quarterly and monthly tourism, Australian labour, Wikipedia article visits, and Bay Area Traffic—showed consistent improvements in CRPS accuracy, averaging over 15 percent. We observed a 4 percent performance drop on the smaller Australian labour dataset. Ablation studies further confirmed the importance of the CRPS learning objective and the enhancements made to support multivariate time series inputs, which were key to the model's success.

Our model's success on datasets with strong Granger causality is due to its vector autoregressive capabilities. However, in the presence of cointegrated series, vector autoregresive approaches can be ineffective. Developing multivariate time series methods that are robust to cointegration is a promising area for future research.

This study uses the reparameterization trick to enable alternative learning objectives such as CRPS and the energy score for neural forecasting tasks and the use of coherent factor models to capture correlations among hierarchical series structures. While we focus on parameterizing the multivariate predictive distribution as a Gaussian multivariate factor model, the framework's flexibility can also accommodate other distributions that support sample differentiability. This is of special interest for outlier quantiles that cannot be well approximated by Gaussian variables. Exciting future research directions include extending the reparameterization trick to handle discrete distributions, which could further enhance the accuracy of forecast distributions built on this framework. Another promising direction for future research is to extend the usage of the reparameterization trick from the learning objectives, into the hierarchical aggregation structure itself, provided the aggregation structure is done through differentiable transformations.

Finally, we observed performance challenges of neural forecasting models when working with limited dataset sizes. Traditional tools like data augmentation and pretraining need to be adapted to handle multivariate time series and associated covariates. Extending these techniques to the hierarchical forecasting domain is a promising research direction.

## Acknowledgements

This work was supported by the Amazon Supply Chain Optimization (SCOT) Forecasting Team. We thank the TMLR reviewers and the Action Editor for their helpful feedbacks and suggestions. We express our gratitude to Stefania La Vattiata for her contribution to the illustrations, which effectively captured the essence of our methods. Special thanks to David Luo for his work on hierarchical forecast baselines. Thanks to Boris Oreshkin for his pointers to latest developments in graph neural networks, and recommendations to enhance our ablation studies. We also appreciate Riccardo Savorgnan and the regional team for enriching discussions on alternatives for adapting the MQCNN architecture to multivariate time series. The authors thank Utkarsh for his suggestion on optimizing the closed-form CRPS for truncated normal variables. We thank Youxin Zhang for his ideas to improve the computational complexity of the Energy Score and the CRPS estimators. Thanks to Lee Dicker and Medha Agarwal for their input on the energy score part of the paper and qualitative evaluation of the forecast calibration.

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

# A Multivariate Factor Model Coherence and Covariance

## A.1 Coherent Aggregation Properties

The coherence of `CLOVER` is a special case of the bootstrap sample reconciliation technique (Panagiotelis et al., 2023), as explored by Olivares et al. (2023).

**Lemma A.1.** *Let $(\Omega_{[b]}, \mathcal{F}_{[b]}, \mathbb{P}_{[b]})$ be a probabilistic forecast space with $\mathcal{F}_{[b]}$ a $\sigma$-algebra in $\Omega_{[b]}$. If a forecast distribution $\mathbb{P}_{[i]}$ assigns a zero probability to sets that do not contain coherent forecasts, it defines a coherent probabilistic forecast space $(\Omega_{[i]}, \mathcal{F}_{[i]}, \mathbb{P}_{[i]})$ with $\Omega_{[i]} = \mathbf{S}_{[i][b]}(\Omega_{[b]})$.*

$$\mathbb{P}_{[a]}\left(\mathbf{y}_{[a]} \notin \mathbf{A}_{[a][b]}(\mathcal{B}) \mid \mathcal{B}\right) = 0 \implies \mathbb{P}_{[i]}\left(\mathbf{S}_{[i][b]}(\mathcal{B})\right) = \mathbb{P}_{[b]}(\mathcal{B}) \quad \forall \mathcal{B} \in \mathcal{F}_{[b]}. \tag{20}$$

*Proof.* We note the following:

$$\mathbb{P}_{[i]}\left(\mathbf{S}_{[i][b]}(\mathcal{B})\right) = \mathbb{P}_{[i]}\left(\begin{bmatrix}\mathbf{A}_{[a][b]} \\ \mathbf{I}_{[b][b]}\end{bmatrix}(\mathcal{B})\right) = \mathbb{P}_{[i]}\left(\{\begin{bmatrix}\mathbf{A}_{[a][b]}(\mathcal{B}) \\ \mathbb{R}^{N_b}\end{bmatrix}\} \cap \{\begin{bmatrix}\mathbb{R}^{N_a} \\ \mathcal{B}\end{bmatrix}\}\right)$$
$$= \mathbb{P}_{[a]}\left(\mathbf{A}_{[a][b]}(\mathcal{B}) \mid \mathcal{B}\right) \mathbb{P}_{[b]}(\mathcal{B}) = (1 - \mathbb{P}_{[a]}\left(\mathbf{y}_{[a]} \notin \mathbf{A}_{[a][b]}(\mathcal{B}) \mid \mathcal{B}\right)) \times \mathbb{P}_{[b]}(\mathcal{B}) = \mathbb{P}_{[b]}(\mathcal{B}).$$

The first equality is the image of a set $\mathcal{B} \in \Omega_{[b]}$ corresponding to the constraints matrix transformation, the second equality defines the spanned space as a subspace intersection of the aggregate series and the bottom series, the third equality uses the conditional probability multiplication rule, and the final equality uses the zero probability assumption.

By construction of the samples of our model $\tilde{\mathbf{y}}_{[i]} = \mathbf{S}_{[i][b]}\left(\hat{\mathbf{y}}_{[b]}\right)_+$ and $\tilde{\mathbf{y}}_{[a]} = \mathbf{A}_{[a][b]}\left(\hat{\mathbf{y}}_{[b]}\right)_+$, satisfying the assumptions of the lemma and proving the coherence of our approach.

$\square$

## A.2 Covariance Structure

Here we prove the covariance structure of our factor model introduced in Section 3.1.

**Lemma A.2.** *Let our factor model be defined by*

$$\hat{\mathbf{y}}_{[b],\eta,t} = \hat{\boldsymbol{\mu}}_{[b],\eta,t} + \mathrm{Diag}(\hat{\boldsymbol{\sigma}}_{[b],\eta,t})\mathbf{z}_{[b],\eta,t} + \hat{\mathbf{F}}_{[b][k],\eta,t}\boldsymbol{\epsilon}_{[k],\eta,t}, \qquad \eta = 1, \cdots, N_h, \tag{21}$$

*with independent factors $\mathbf{z}_{[b],\eta} \sim \mathcal{N}(\mathbf{0}_{[b]}, \mathbf{I}_{[b][b]})$, and $\boldsymbol{\epsilon}_{[k],\eta} \sim \mathcal{N}(\mathbf{0}_{[k]}, \mathbf{I}_{[k][k]})$, its covariance satisfies*

$$\mathrm{Cov}\left(\hat{\mathbf{y}}_{[b],\eta,t}\right) = \mathrm{Diag}(\hat{\boldsymbol{\sigma}}_{[b],\eta,t}^2) + \hat{\mathbf{F}}_{[b][k],\eta,t}\hat{\mathbf{F}}_{[b][k],\eta,t}^\top. \tag{22}$$

*Proof.* First, we observe that

$$\begin{aligned}
\mathrm{Cov}\left(\hat{\mathbf{y}}_{[b],\eta,t}, \hat{\mathbf{y}}_{[b],\eta,t}\right) &= \mathrm{Cov}\left(\mathrm{Diag}(\hat{\boldsymbol{\sigma}}_{[b],\eta,t})\mathbf{z}_{[b],\eta,t}, \mathrm{Diag}(\hat{\boldsymbol{\sigma}}_{[b],\eta,t})\mathbf{z}_{[b],\eta,t}\right) \\
&+ 2\mathrm{Cov}\left(\mathrm{Diag}(\hat{\boldsymbol{\sigma}}_{[b],\eta,t})\mathbf{z}_{[b],\eta,t}, \hat{\mathbf{F}}_{[b][k],\eta,t}\boldsymbol{\epsilon}_{[k],\eta,t}\right) \\
&+ \mathrm{Cov}\left(\hat{\mathbf{F}}_{[b][k],\eta,t}\boldsymbol{\epsilon}_{[k],\eta,t}, \hat{\mathbf{F}}_{[b][k],\eta,t}\boldsymbol{\epsilon}_{[k],\eta,t}\right).
\end{aligned} \tag{23}$$

By bilinearity of covariance and independence of the sampled factors, it follows that

$$\mathrm{Cov}\left(\hat{\mathbf{y}}_{[b],\eta,t}, \hat{\mathbf{y}}_{[b],\eta,t}\right) = \mathrm{Diag}(\hat{\boldsymbol{\sigma}}_{[b],\eta,t})\mathrm{Cov}\left(\mathbf{z}_{[b],\eta,t}, \mathbf{z}_{[b],\eta,t}\right)\mathrm{Diag}(\hat{\boldsymbol{\sigma}}_{[b],\eta,t})^\top + \hat{\mathbf{F}}_{[b][k],\eta,t}\mathrm{Cov}\left(\boldsymbol{\epsilon}_{[k],\eta,t}, \boldsymbol{\epsilon}_{[k],\eta,t}\right)\hat{\mathbf{F}}_{[b][k],\eta,t}^\top.$$

We conclude that

$$\mathrm{Cov}\left(\hat{\mathbf{y}}_{[b],\eta,t}, \hat{\mathbf{y}}_{[b],\eta,t}\right) = \mathrm{Diag}(\hat{\boldsymbol{\sigma}}_{[b],\eta,t}^2) + \hat{\mathbf{F}}_{[b][k],\eta,t}\hat{\mathbf{F}}_{[b][k],\eta,t}^\top. \tag{24}$$

$\square$

## B  Code Script for Sampling

```python
def sample(self, distr_args, window_size, num_samples=None):
    """
    **Parameters**
    'distr_args': Forecast Distribution arguments.
    'window_size': int=1, for reconciliation reshapes in sample method.
    'num_samples': int=500, number of samples for the empirical quantiles.

    **Returns**
    'samples': tensor, shape [B,H,'num_samples'].
    'quantiles': tensor, empirical quantiles defined by 'levels'.
    """
    means, factor_loading, stds = distr_args
    collapsed_batch, H, _ = means.size()

    # [collapsed_batch,H]:=[B*N*Ws,H,F] -> [B,N,Ws,H,F]
    factor_loading = factor_loading.reshape(
        (-1, self.n_series, window_size, H, self.n_factors)
    ).contiguous()
    factor_loading = torch.einsum("iv,bvwhf->biwhf", self.SP, factor_loading)

    means = means.reshape(-1, self.n_series, window_size, H, 1).contiguous()
    stds = stds.reshape(-1, self.n_series, window_size, H, 1).contiguous()

    # Loading factors for covariance Diag(stds) + F F^t -> (SPF)(SPF^t)
    hidden_factor = Normal(
        loc=torch.zeros(
            (factor_loading.shape[0], window_size, H, self.n_factors),
        scale=1.0)
    sample_factors = hidden_factor.rsample(sample_shape=(self.num_samples,))
    sample_factors = sample_factors.permute(
        (1, 2, 3, 4, 0)
    ).contiguous()  # [n_items, window_size, H, F,num_samples]

    sample_loaded_factors = torch.einsum("bvwhf,bwhfn->bvwhn", factor_loading,
                                                              sample_factors)
    sample_loaded_means = means + sample_loaded_factors

    # Sample Normal
    normal = Normal(loc=torch.zeros_like(sample_loaded_means), scale=1.0)
    samples = normal.rsample()
    samples = F.relu(sample_loaded_means + stds * samples)

    samples = torch.einsum("iv,bvwhn->biwhn", self.SP, samples)
    samples = samples.reshape(collapsed_batch, H, self.num_samples).contiguous()

    # Compute quantiles and mean
    quantiles_device = self.quantiles.to(means.device)
    quants = torch.quantile(input=samples, q=quantiles_device, dim=-1)
    quants = quants.permute((1, 2, 0))  # [Q,B,H] -> [B,H,Q]
    sample_mean = torch.mean(samples, dim=-1, keepdim=True)
    return samples, sample_mean, quants
```

Figure 4: PyTorch function for sampling from our Gaussian Factor model. Note that the factor samples are shared across all bottom-level distributions. The samples are differentiable with regard to the function inputs. We can easily adapt this function to sample from other distributions.

Table 4: *Coherent Learning Objective Reparameterization Neural Network* (`CLOVER`) hyperparameters. We use a small or large model configuration depending on the datasets' size.

| PARAMETER | Considered Values | | | | | |
| --- | --- | --- | --- | --- | --- | --- |
| | TRAFFIC | TOURISM-S | WIKI | LABOUR | TOURISM-L | FAVORITA |
| Activation Function. | ReLU | ReLU | ReLU | ReLU | ReLU | ReLU |
| Static Encoder Dimension. | 5 | 10 | 5 | 10 | 20 | 20 |
| Temporal Convolution Channel Size. | 10 | 10 | 10 | 20 | 30 | 30 |
| Future Encoder Dimension. | 20 | 4 | 20 | 20 | 50 | 50 |
| Horizon Specific Decoder Dimensions. | 5 | 5 | 5 | 5 | 5 | 5 |
| Horizon Agnostic Decoder Dimensions. | 20 | 10 | 20 | 10 | 20 | 20 |
| Factor Model Components. | 10 | 10 | 20 | 20 | 10 | 5 |
| Cross Series MLP Hidden Size. | 200 | 100 | 200 | 0 | 50 | 5 |
| SGD Batch Size. | 1 | 1 | 1 | 1 | 1 | 4 |
| SGD Effective Batch Size. | 207 | 89 | 199 | 57 | 555 | 744 |
| SGD Max steps. | 2e3 | 1e3 | 2e3 | 2e3 | 2e3 | 80e3 |
| Early Stop Patience steps. | 5 | 5 | 5 | 4 | 5 | -1 |
| Learning Rate. | 5e-3 | 5e-4 | 5e-4 | 1e-3 | 5e-4 | 5e-4 |

## C   Training Methodology and Hyperparameters

Here, we complement and extend the description of our method in Section 3.

To avoid information leakage, we perform ablation studies in the validation set preceding the test set, where we explored variants of the probabilistic method, as well as its optimization. We report these ablation studies in the Appendix F. For each dataset, given the prediction horizon $h$, the test set is composed of the last $h$ time steps. The validation set is composed of the $h$ time-steps preceding the test set time range. The training set is composed of all dates previous to the validation time-range. When reporting the final accuracy results of our model in the test set, we used the settings that perform the best in the validation set.

We minimally tune the architecture and its parameters varying only its size and the convolution kernel filters to match the seasonalities present in each dataset. For the data set `Favorita`, we use dilations of $[1, 2, 4, 8, 16, 32]$ to match the weekly and monthly seasonalities. For the `Tourism-L` and `Labour` datasets we use dilations of $[1, 2, 3, 6, 12]$ to match the monthly and yearly seasonalities. For the `Traffic` dataset we use dilations of $[1, 7, 14, 28]$ as multiples of 7 to match the weekly seasonalities. For the `Tourism-S` dataset we use dilations of $[1, 2, 4]$ as multiples of 4 to match the quarterly seasonalities. For the `Wiki` dataset we use minimal dilations $[1, 2]$.

The selection of the number of factors mostly follows the memory constraints of the GPU, as the effective batch size implied by our probabilistic model grows rapidly as a function of the multivariate series. In the `Favorita` dataset, more factors are likely to continue to improve accuracy but with the tradeoff of the computational speed. Similarly, the Cross-series MLP hidden size is selected following the GPU memory constraints.

We share a learning rate of 5e-4 constant across the three datasets, which shows that the method is reasonably robust across different forecasting tasks. During the optimization of the networks we use adaptive moment stochastic gradient descent (Kingma & Ba, 2014) with early stopping (Yao et al., 2007) guided by the sCRPS signal measured in the validation set. We use a learning rate scheduler that decimates the learning rate four times during optimization (SGD Maxsteps / 4), to ensure the convergence of the optimization.

The `CLOVER` model is implemented using PyTorch (Paszke et al., 2019), with the NeuralForecast library framework (Olivares et al., 2022b). We run all experiments using a single NVIDIA V100 GPU.

As mentioned earlier in Section 4.1, statistical methods available in StatsForecast and HierarchicalForecast libraries (Olivares et al., 2022c; Garza et al., 2022). In particular, we use available code on the hierarchical baselines repository and hierarchical datasets repository. As described in Table 2, we deviate slightly from the experimental setting in Rangapuram et al. (2021) to ensure the reproducibility of the results of this paper. For Table 3 we rerun the `HierE2E` (Rangapuram et al., 2021) baseline on `Labour`, `Traffic`, `Wiki` and `Tourism-S` using their best reported hyperparameters.

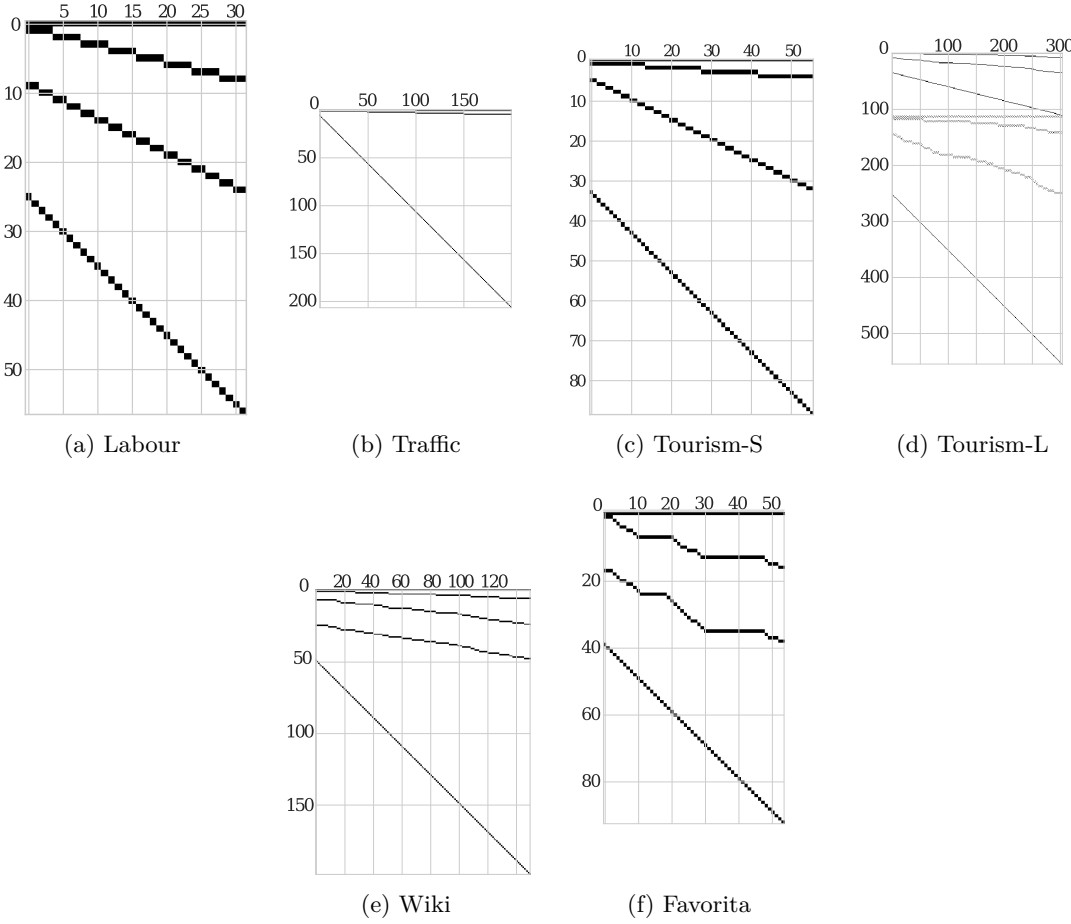

Figure 5: Hierarchical constraints of the empirical evaluation datasets. (a) `Labour` reports 57 series number of employees by full-time status, gender and geographic levels. (b) `Traffic` organizes the occupancy series of 200 highways into quarters, halves, and totals. (c) `Tourism-S` categorizes its 89 regional visit series based on travel purpose, zones, states, and country-level aggregations and urbanization within regions. (d) `Tourism-L` categorizes its 555 regional visit series based on travel purpose, zones, states, and country-level geographical aggregations. (e) `Wiki` groups 150 daily visits to Wikipedia articles by language and article categorical taxonomy. (f) `Favorita` classifies its grocery sales by store, city, state, and country levels.

# D   Dataset Details

`Labour:`   The Labour dataset (Australian Bureau of Statistics, 2019) tracks monthly Australian employment from February 1978 to December 2019, reporting total employees by part-time/full-time status, gender, and geography. It includes $N = 57$ series in total, with $N_a = 25$ aggregate series and $N_b = 32$ bottom-level series. We use 8 months from May 2019 to December 2019 as test, and the rest of the data as training and validation.

`Traffic:`   The Traffic dataset (Ben Taieb & Koo, 2019) contains daily (aggregated from hourly rates) freeway occupancy rates for 200 car lanes in the San Francisco Bay Area, aggregated from January 2008 to March 2009. The data is grouped into three levels: four groups of 50 lanes, two groups of 100 lanes, and one overall group of 200 lanes, following the random grouping in Rangapuram et al. (2021); Olivares et al. (2023). Consistent with prior work (Ben Taieb & Koo, 2019; Rangapuram et al., 2021; Olivares et al., 2023), we split the dataset into 120 training, 120 validation, and 126 test samples, reporting accuracy for the last test date. We use geographic node dummies, weekend indicators, and proximity to Saturday for exogenous variables.

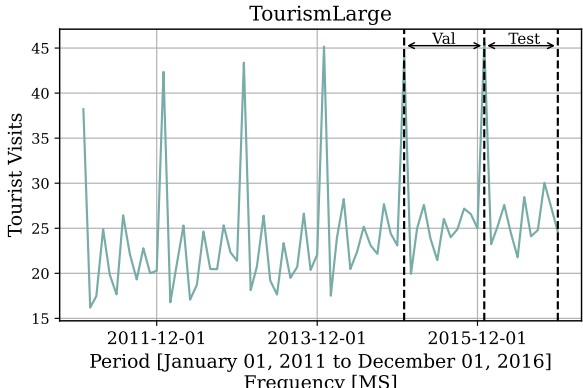

Figure 6: `Tourism-L` dataset partition into train, validation, and test sets used in our experiments. All datasets use the last horizon window as defined in Table 2 (marked by the second dotted line), and the previous window preceding the test set as validation (between the first and second dotted lines). Validation provides the signal for hyperparameter selection and the ablation studies.

**Tourism-S:** The `Tourism-S` dataset (Tourism Australia, Canberra, 2005) records quarterly visits to Australia from 1996 to 2006. We use data from 2005 for validation, 2006 for testing, and the remaining quarters for training. Each series contains 28 quarterly observations and is structured with aggregated visit data by country, purpose of travel, state, regions, and urbanization level within regions. The most disaggregated level includes 56 regions by urbanization, while the aggregated data consists of 33 series. We use state dummies, and for the future exogenous variables, we use quarterly dummies and seasonal naive 4 and 8 anchors.

**Tourism-L:** The `Tourism-L` dataset (Wickramasuriya et al., 2019) represents visits to Australia, at a monthly frequency, between January 1998 and December 2016. We use 2015 for validation, and 2016 for testing, and all previous years for training. The dataset contains 228 monthly observations. For each month, we have the number of visits to each of Australia's 78 regions, which are aggregated to the zone, state, and national level, and for each of four purposes of travel. These two dimensions of aggregation total $N = 304$ leaf entities (a region-purpose pair), with a total of $M = 555$ series in the hierarchy.

We pre-process the data to include static features. We use purpose of travel as well as state dummies. For the historical information, we use month dummies, and for the future exogenous variables, we use month dummies and a seasonal naive anchor forecast that helps greatly to account for the series seasonality.

**Favorita:** The Favorita dataset (Favorita et al., 2017) contains grocery sales of the Ecuadorian Corporación Favorita in $N = 54$ stores. We perform geographical aggregation of the sales at the store, city, state, and national levels, following (Olivares et al., 2023). This yields a total of $M = 94$ aggregates. Concerning features, we use past unit sales and number of transactions as historical data. In the `Favorita` dataset, we include item perishability static information, geographic state dummy variables, and for the historic exogenous features and future exogenous features, we use promotions and day of the week. During the model's optimization we consider a balanced dataset of items and stores, for 217,944 bottom level series (4,036 items * 54 stores), along with aggregate levels for a total of 371,312 time series. The dataset is at the daily level and starts from 2013-01-01 and ends on 2017-08-15, comprehending 1688 days. We keep 34 days (1654 to 1988 days) as hold-out test and 34 days (1620 to 1654 days) as validation.

**Wiki:** The Wikipedia dataset (Anava et al., 2018) contains daily views of 145,000 online articles from July 2015 to December 2016. The dataset is processed into $N_b = 150$ bottom series and $N_a = 49$ aggregate series based on country, access, agent and article categories, following Ben Taieb & Koo (2019); Rangapuram et al. (2021) processing. The last week of December 2016 is used as test, and the remaining data as training and validation. We use day of the week dummies to capture seasonalities, country, access, agent dummy static features, and a seasonal naive 7 anchor.

# E    Forecast Distributions Visualization

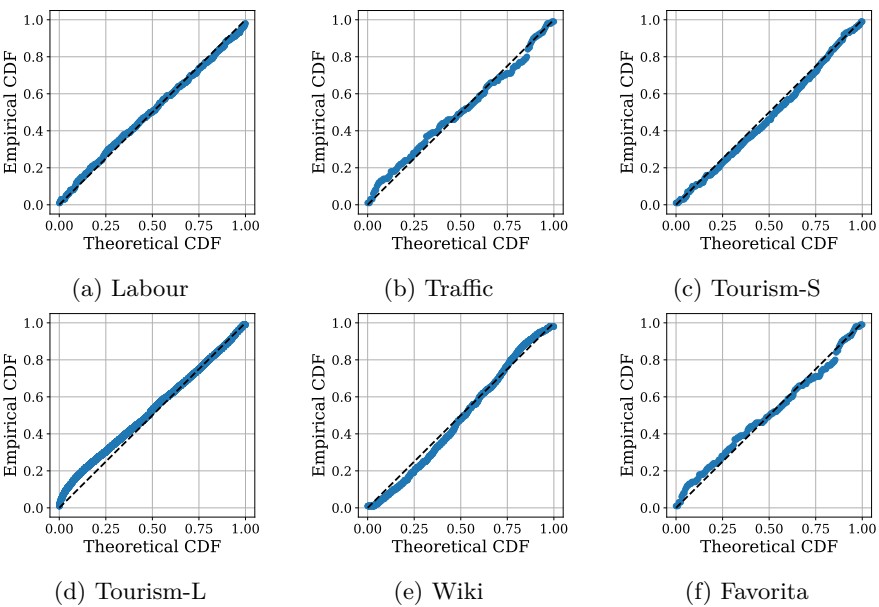

Figure 7: Probability-Probability plot comparing the cumulative probabilities of the empirical distribution and the CLOVER' forecast distribution. Points near the 45-degree line indicate similarity between the distributions. The plot uses a single run from the models reported in the main results Table 3.

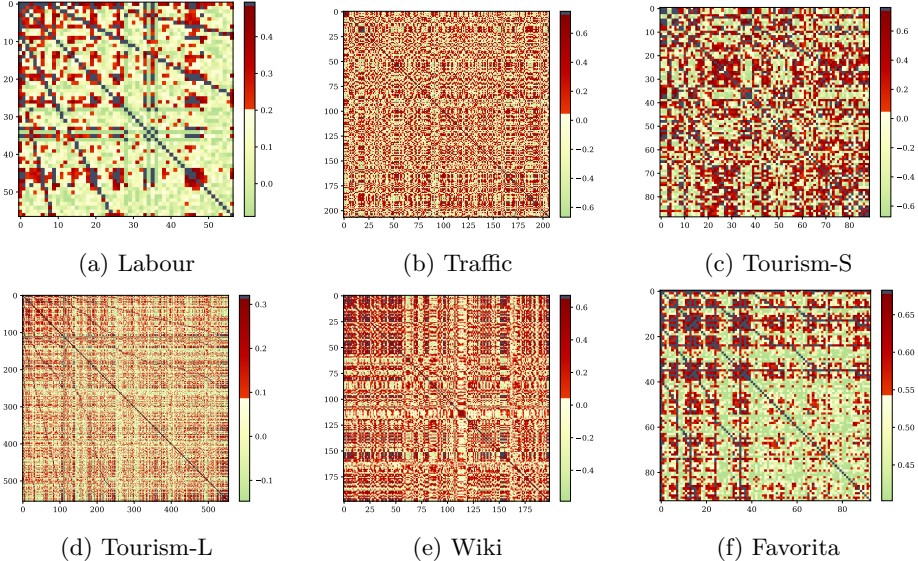

Figure 8: Estimated correlation matrices produced by CLOVER. The color scale indicates the strength of correlation between pairs of series in the hierarchy. Green represents negative correlations, red indicates positive correlations, and black highlights the most highly correlated series. The estimated covariances are sparse, with many correlations close to zero. Aggregated levels, seen in the top-left corner, show stronger positive correlations. The plot uses a single run from the models in Table 3.

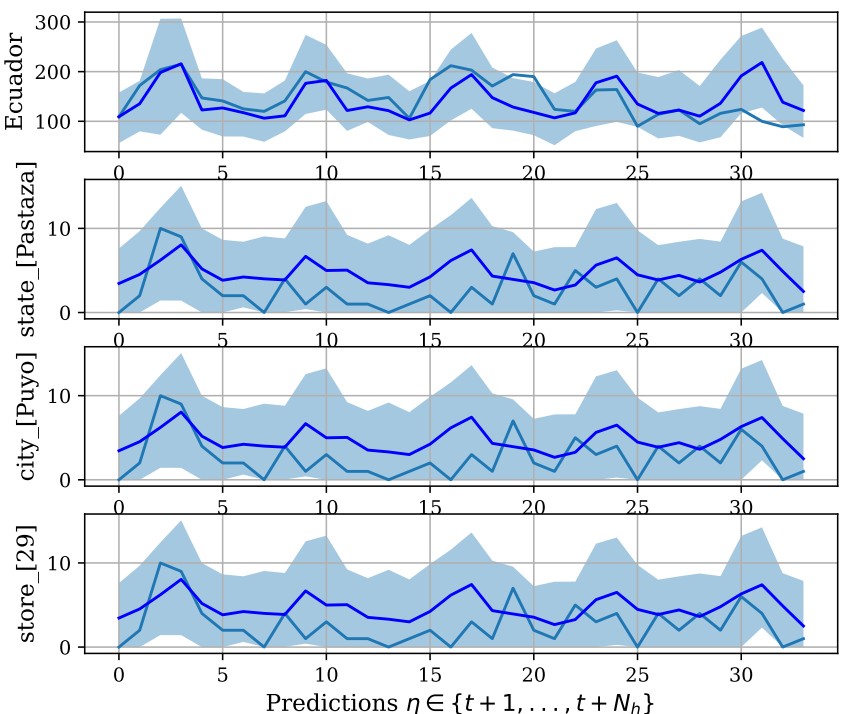

Figure 9: `CLOVER` forecast distributions on the `Favorita` dataset. We show the forecasted demand for a grocery item on a store of the Puyo City, in the State of Pastaza and the whole country demand in the top row. Forecast distributions show the 90% forecast intervals in light blue, and the forecasted median in dark blue. The clipped Normal distribution achieves non-negative predictions and a point mass at zero.

Table 5: Ablation study on the `Traffic` dataset, empirical evaluation of probabilistic coherent forecasts. Mean *scaled continuous ranked probability score* (sCRPS) averaged over 5 runs, at each aggregation level, the best result is highlighted (lower measurements are preferred). We report 95% confidence intervals. Comparison of different neural forecasting architectures augmented with the Factor Model and trained with CRPS.

| Level | CLOVER | MQCNN | TFT | NHITS | NBEATS | LSTM | FCGAGA |
|---|---|---|---|---|---|---|---|
| Overall | 0.0242±0.0035 | 0.0613±0.0257 | 0.0698±0.0106 | 0.0644±0.0095 | 0.0688±0.0156 | 0.0646±0.0066 | 0.0592±0.0235 |
| Total | 0.0035±0.0015 | 0.0432±0.0296 | 0.0311±0.0185 | 0.0339±0.0131 | 0.0423±0.0175 | 0.0377±0.0130 | 0.0241±0.0103 |
| Halves | 0.0048±0.0030 | 0.0437±0.0289 | 0.0402±0.0104 | 0.0376±0.0122 | 0.0443±0.0159 | 0.0402±0.0089 | 0.0259±0.0099 |
| Quarters | 0.0041±0.0022 | 0.0440±0.0281 | 0.0441±0.0111 | 0.0420±0.0084 | 0.0478±0.0132 | 0.0402±0.0084 | 0.0285±0.0083 |
| Lanes | 0.0905±0.0084 | 0.1145±0.0174 | 0.1686±0.0159 | 0.1442±0.0213 | 0.1409±0.0182 | 0.1404±0.0071 | 0.1583±0.0923 |

Table 6: Ablation study on the `Traffic` dataset. empirical evaluation of probabilistic coherent forecasts. Mean *scaled continuous ranked probability score* (sCRPS) averaged over 5 runs, at each aggregation level, the best result is highlighted (lower measurements are preferred). We report 95% confidence intervals. Comparison of `MQCNN`-based architecture trained with different learning objectives.

[*] The Normal and StudentT are non coherent forecast distributions, in contrast to the Factor Model and the Poisson Mixture.

| Level | CLOVER | | | Other Distributions | | |
| | CRPS | Energy Score | NLL | PoissonMixture | StudentT* | Normal* |
|---|---|---|---|---|---|---|
| Overall | 0.0259±0.0060 | 0.0269±0.0044 | 0.0879±0.1136 | 0.0827±0.1408 | 0.0600±0.0367 | 0.0734±0.0253 |
| Total | 0.0023±0.0022 | 0.0031±0.0030 | 0.0623±0.1335 | 0.0667±0.1817 | 0.0280±0.0283 | 0.0556±0.0327 |
| Halves | 0.0028±0.0018 | 0.0035±0.0027 | 0.0640±0.1318 | 0.0765±0.2030 | 0.0296±0.0300 | 0.0510±0.0316 |
| Quarters | 0.0043±0.0015 | 0.0074±0.0019 | 0.0644±0.1313 | 0.0742±0.1782 | 0.0301±0.0294 | 0.0450±0.0334 |
| Lanes | 0.0942±0.0195 | 0.0882±0.0159 | 0.1608±0.0615 | 0.1136±0.0056 | 0.1523±0.0752 | 0.1422±0.0554 |

## F Ablation Studies Details

To analyze the sources of improvements in our model, we conducted ablation studies on variants of the `CLOVER`/`MQCNN`/`DPMN` (Wen et al., 2017; Olivares et al., 2022c). We used a simplified setup on the `Traffic` dataset, focusing on the same forecasting task as in the main experiment. We evaluated sCRPS from Eqn. 19 in the validation set across five randomly initialized neural networks. The experiments Table 4 hyperparameters, and vary a single characteristic of the network, and measure its validation effects.

In our first ablation study, we explore the effects of the impact of including vector autoregressive relationships of the hierarchy through the CrossSeriesMLP module described in Eqn. 14. In the experiment, we train a `CLOVER` with and without the module in the `Traffic` dataset. Additionally we compare different well-performing neural forecasting architectures[6] augmented with the CrossSeriesMLP including (1) `LSTM` (Sak et al., 2014), (2) `NBEATS` (Oreshkin et al., 2020; Olivares et al., 2022a), (3) `NHITS` (Challu et al., 2023), (4) `TFT` (Lim et al., 2021), (5) `FCGAGA` (Oreshkin et al., 2021) a network specialized in spatio-temporal forecasting. We use the default implementations available in the NeuralForecast library (Olivares et al., 2022b) [7]. Table 6 shows that convolution-based architectures continue to deliver state-of-the-art results., more importantly using the CrossSeriesMLP improves sCRPS upon the alternative (without) by 66 percent. The technique bridges the gap to the `HierE2E` (Rangapuram et al., 2021), which previously outperformed all alternative methods by over 50 percent. We attribute the improvements to the heavy presence of Granger causal relationships between the traffic lanes, as they carry lag historical information that influences each other.

In our second ablation study, we explore learning objective alternatives to Eqn. 17. For this purpose, we replace the last layer of `CLOVER` with different distribution outputs, including the normal, Student-t, and Poisson mixture distributions (Olivares et al., 2023). In addition we also compare with our own Factor model approach, as we can see in Table 6 and Figure 3, the CRPS optimization of the Factor Model improves upon the negative log likelihood by 60 percent in the mean sCRPS in the validation set. The difference is highly driven by outlier runs, but it is expected as the CRPS objective has much more convenient numerical properties, starting with its bounded gradients.

---

[6]The Factor Model can augment models from the Neural Forecast library (Olivares et al., 2022b). Because of this, it can readily augment AutoFormer, BitCN, DeepAR, DeepNPTS, DilatedRNN, DLinear, FedFormer, FCGaga, GRU, Informer, ITransformer, KAN, LSTM, MLP, NBEATS, NBEATSx, NHITS, NLinear, PatchTST, RMOK, RNN, SOFTS, TCN, TFT, Tide, TimeLLM, TimeMixer, and TimesNet. For this ablation study, we focus on NBEATS, NHITS, LSTM, FCGaga, and TFT.

[7]Optimization of the neural forecasting networks follows details from Appendix C, we increased four times the early stopping patience on `NHITS`/`NBEATS`/`FCGAGA` to account for the gradient variance compared to `MQCNN`'s gradient based on forking sequences.

Table 7: Empirical evaluation of mean hierarchical forecasts. *Relative squared error* (relSE) averaged over 5 runs, at each aggregation level, the best result is highlighted (lower values are preferred). We report 95% confidence intervals, the methods without standard deviation have deterministic solutions.
[*] The ARIMA-ERM results for Tourism-L differ from Rangapuram et al. (2021), as we improved the numerical stability of their implementation.

| DATA | LEVEL | CLOVER (crps) | CLOVER (energy) | DPMN-GroupBU | ARIMA-ERM [*] | ARIMA-MinT-ols | ARIMA-BU | ARIMA (not hier.) | SNaive (not hier.) |
|---|---|---|---|---|---|---|---|---|---|
| Labour | Overall | 0.3882±0.0507 | 0.3816±0.0813 | 0.7896±0.2825 | 0.5671 | **0.3136** | 0.4527 | 0.3289 | 5.0683 |
| | 1 (geo.) | 0.1483±0.0614 | 0.1421±0.0902 | 0.3307±0.3012 | 0.0395 | **0.0320** | 0.1455 | 0.0369 | 5.9572 |
| | 2 (geo.) | 0.4965±0.0898 | 0.4889±0.1322 | 1.3206±0.4084 | 0.4274 | **0.3590** | 0.5927 | 0.3609 | 5.8649 |
| | 3 (geo.) | 0.6645±0.0528 | **0.6580±0.0765** | 1.2675±0.2874 | 0.9561 | 0.6807 | 0.8742 | 0.6723 | 4.0696 |
| | 4 (geo.) | 0.7301±0.0236 | **0.7234±0.0285** | 1.1693±0.3414 | 1.8568 | 0.7605 | 0.8364 | 0.8364 | 2.6208 |
| Traffic | Overall | **0.0008±0.0004** | 0.0021±0.0031 | 0.1750±0.0099 | 0.0199 | 0.0425 | 0.0217 | 0.0433 | 0.0709 |
| | 1 (geo.) | **0.0001±0.0002** | 0.0013±0.0027 | 0.1619±0.0099 | 0.0133 | 0.0344 | 0.0168 | 0.0302 | 0.0547 |
| | 2 (geo.) | **0.0001±0.0003** | 0.0014±0.0031 | 0.1835±0.0101 | 0.0135 | 0.0380 | 0.0180 | 0.0392 | 0.0676 |
| | 3 (geo.) | **0.0005±0.0007** | 0.0017±0.0035 | 0.1819±0.0100 | 0.0373 | 0.0647 | 0.0295 | 0.0850 | 0.0989 |
| | 4 (geo.) | **0.1354±0.0325** | 0.1454±0.0604 | 0.9964±0.0430 | 0.6355 | 0.5876 | 0.5669 | 0.5669 | 1.3118 |
| Tourism-S | Overall | 0.1078±0.0129 | 0.1104±0.0131 | 0.1141±0.0241 | 0.3781 | 0.1772 | **0.0994** | 0.1886 | 0.2198 |
| | 1 (geo.) | 0.1044±0.0189 | 0.0874±0.0202 | 0.1362±0.0359 | 0.4174 | 0.1722 | **0.0696** | 0.1649 | 0.2596 |
| | 2 (geo.) | 0.0969±0.0060 | 0.1004±0.0175 | 0.0797±0.0103 | 0.367 | 0.1605 | **0.0819** | 0.2066 | 0.1741 |
| | 3 (geo.) | **0.1381±0.0098** | 0.1642±0.0097 | 0.1241±0.0299 | 0.3033 | 0.2015 | 0.1684 | 0.1819 | 0.2163 |
| | 4 (geo.) | **0.1576±0.0079** | 0.1872±0.0206 | 0.1597±0.0371 | 0.3369 | 0.2444 | 0.2218 | 0.2218 | 0.2557 |
| Tourism-L | Overall | **0.0951±0.0145** | 0.0952±0.0188 | 0.1113±0.0158 | 0.1178 | 0.1251 | 0.2979 | 0.1414 | 0.1306 |
| | 1 (geo.) | 0.0447±0.0171 | 0.0405±0.0327 | 0.0597±0.0212 | 0.0596 | 0.0472 | 0.4002 | **0.0343** | 0.0582 |
| | 2 (geo.) | **0.1014±0.0180** | 0.1016±0.0304 | 0.1121±0.0152 | 0.1293 | 0.1476 | 0.3340 | 0.2530 | 0.1628 |
| | 3 (geo.) | 0.2309±0.0124 | 0.2299±0.0253 | **0.2250±0.0196** | 0.2529 | 0.3556 | 0.4238 | 0.4429 | 0.3695 |
| | 4 (geo.) | 0.3075±0.0134 | 0.3263±0.0296 | **0.2980±0.0197** | 0.3236 | 0.4288 | 0.4012 | 0.4835 | 0.4766 |
| | 5 (prp.) | 0.0596±0.0195 | **0.0560±0.0078** | 0.0798±0.0195 | 0.0895 | 0.0856 | 0.1703 | 0.0973 | 0.0615 |
| | 6 (prp.) | **0.1199±0.0115** | 0.1232±0.0085 | 0.1403±0.0150 | 0.1466 | 0.1537 | 0.1986 | 0.1663 | 0.1577 |
| | 7 (prp.) | **0.2484±0.0119** | 0.2644±0.0195 | 0.2654±0.0212 | 0.2705 | 0.3017 | 0.3151 | 0.2914 | 0.3699 |
| | 8 (prp.) | 0.3432±0.0157 | 0.3830±0.0260 | **0.3302±0.0235** | 0.3543 | 0.3970 | 0.3769 | 0.3769 | 0.4969 |
| Wiki | Overall | **0.5301±0.0502** | 0.7639±0.1517 | 0.9270±0.1237 | 0.9768 | 1.0253 | 0.9621 | 1.0419 | 0.9288 |
| | 1 (geo.) | **0.1980±0.0626** | 0.4673±0.2430 | 0.6294±0.1769 | 0.8347 | 1.0528 | 0.8271 | 1.1357 | 0.6555 |
| | 2 (geo.) | **0.4963±0.0548** | 0.7627±0.1917 | 0.9266±0.1056 | 1.0342 | 0.9913 | 0.9615 | 0.8300 | 1.0672 |
| | 3 (geo.) | **0.8272±0.0777** | 0.9492±0.3670 | 1.2059±0.0997 | 1.0844 | 1.0097 | 1.1105 | 1.0398 | 1.1441 |
| | 4 (geo.) | **0.8152±0.0732** | 0.9339±0.3571 | 1.1789±0.0961 | 1.0744 | 1.0030 | 1.0858 | 1.0259 | 1.1095 |
| | 5 (prp.) | **0.8820±0.0698** | 1.2234±0.5854 | 1.1958±0.0732 | 1.0881 | 1.0294 | 1.0504 | 1.0504 | 1.1080 |
| Favorita | Overall | **0.5885±0.0291** | 0.7517±0.0255 | 0.7563±0.0713 | 0.8163 | 0.9465 | 0.8276 | 0.9665 | 1.1420 |
| | 1 (geo.) | **0.6109±0.0400** | 0.7864±0.0291 | 0.7944±0.0568 | 0.8362 | 0.8999 | 0.8415 | 0.9217 | 1.1269 |
| | 2 (geo.) | **0.5618±0.0265** | 0.7183±0.0216 | 0.7355±0.1057 | 0.7830 | 1.0057 | 0.8050 | 1.0451 | 1.1078 |
| | 3 (geo.) | **0.5619±0.0256** | 0.7256±0.0259 | 0.7303±0.1035 | 0.7986 | 1.0418 | 0.8192 | 1.0881 | 1.1315 |
| | 4 (geo.) | **0.5854±0.0130** | 0.7083±0.0274 | 0.6770±0.0351 | 0.8199 | 0.8808 | 0.8228 | 0.8228 | 1.2815 |

## G    Mean Forecast Accuracy Evaluation

To complement the probabilistic and L1-based results in Section 4.1. We also evaluate mean forecasts denoted by $\bar{\mathbf{y}}_{[i][h],t} := (\bar{\mathbf{y}}_{[i],1,t}, \cdots, \bar{\mathbf{y}}_{[i],N_h,t})$ through the *relative squared error* relSE (Hyndman & Koehler, 2006), that considers the ratio between squared error across forecasts in all levels over squared error of the Naive forecast (i.e., a point forecast using the last observation $\mathbf{y}_{[i],t}$) as described by

$$\text{relSE}\left(\mathbf{y}_{[i][h],t},\ \bar{\mathbf{y}}_{[i][t+1:t+N_h]} \mid \boldsymbol{l}^{(g)}\right) = \frac{\sum_{i=1}^{N_a+N_b} \left\| \mathbf{y}_{i,[t+1:t+N_h]} - \bar{\mathbf{y}}_{i,[h],t} \right\|_2^2 \cdot l_i^{(g)}}{\sum_{i=1}^{N_a+N_b} \left\| \mathbf{y}_{i,[t+1:t+N_h]} - y_{i,t} \cdot \mathbf{1}_{[h]} \right\|_2^2 \cdot l_i^{(g)}}. \quad (25)$$

As discussed in Section 3.1, our factor model naturally defines a probabilistic coherent system, where hierarchical coherence emerges as a consequence. In this section, we compare our method with the following coherent mean approaches: (1) DPMN-GroupBU, (2) ARIMA-ERM (Ben Taieb & Koo, 2019), (3) ARIMA-MinT (Wickramasuriya et al., 2019), (4) ARIMA-BU, (5) ARIMA, and (6) Seasonal Naive. The statistical methods are implemented using the StatsForecast and HierarchicalForecast libraries (Olivares et al., 2022c; Garza et al., 2022). In particular we use available code on the hierarchical baselines repository, and hierarchical datasets repository.

The CLOVER model improves relSE accuracy in five out of six datasets by an average of 32%, while degrading accuracy on the smallest dataset, Labour, by 32%. These changes can be attributed to squared error metrics' sensitivity to outliers, though the results are generally consistent with sCRPS outcomes.

## H   Efficient Estimation of Scoring Rules

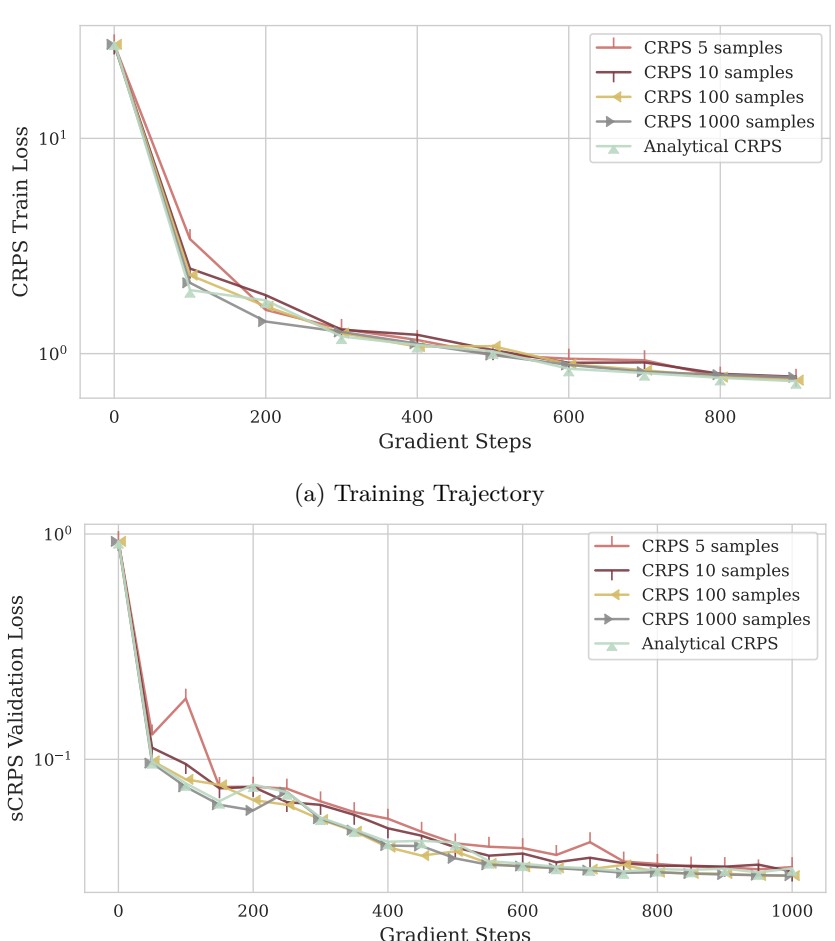

(a) Training Trajectory

(b) Validation Trajectory

Figure 10: Training and validation *Continuous Ranked Probability Score* curves on the SF Bay Area `Traffic` dataset. We show curves for `CLOVER` varying as a function of the gradient steps. Each curve uses a different number of Monte Carlo samples to estimate CRPS.

For the network's optimization we implement the following CRPS Monte Carlo estimator for Eqn. 5

$$\hat{\text{CRPS}}\left(y, Y(\boldsymbol{\theta})\right) = \frac{1}{N}\sum_{i=1}^{N}|y_i - y| - \frac{1}{2N(N-1)}\sum_{i,j=1}^{N}|y_i - y_j| \tag{26}$$

Where $y_i, y_j \quad i,j = 1,...,N$ are independent samples of the random variable $Y(\boldsymbol{\theta})$, and $y$ a possible observation.

To assess the effect of sample size on the CRPS Monte Carlo estimator, we ran diagnostics throughout the model training process. Figure 10 shows the training and validation loss trajectories for varying Monte Carlo sample sizes, using model configurations from Table 4. For reference, we include the analytical CRPS of truncated normals (Thorarinsdottir & Gneiting, 2009), despite minor distributional differences compared to Eqn. 10. For a comprehensive review of alternative CRPS estimators, we refer to Zamo & Naveau (2018).

We found that reducing the number of Monte Carlo samples had minimal impact on both training and validation scores. Given GPU memory constraints, decreasing the sample size effectively reduces computational overhead without significantly affecting model performance. We observed similar phenomena with the energy score learning objective and its relationship to Monte Carlo samples.

# I    Note on the Reparameterization Trick

In this section, we discuss the importance of the reparameterization trick in the context of hierarchical series and our multivariate factor model. We explain how it enables us to maintain the network's differentiability while optimizing arbitrary learning objectives, regardless of the complexity of our probabilistic model.

Let a disaggregated series be

$$\hat{Y}(\boldsymbol{\theta}) = \mu(\boldsymbol{\theta}) + \text{Diag}(\sigma(\boldsymbol{\theta}))\mathbf{z} + \mathbf{F}(\boldsymbol{\theta})\boldsymbol{\epsilon} \in \mathbb{R}^{N_b}. \tag{27}$$

Let the hierarchical series under aggregation constraints be

$$\tilde{Y}(\boldsymbol{\theta}) = \mathbf{S}_{[i][b]}\hat{Y}(\boldsymbol{\theta})_{+}, \tag{28}$$

with $\boldsymbol{\epsilon} \sim N(\mathbf{0}, \mathrm{I}_{N_k})$ and $\mathbf{z} \sim N(\mathbf{0}, \mathrm{I}_{N_b})$ the reparameterization random variables. For simplicity of exposition, we consider a single entry in the reconciled multivariate series $\tilde{Y}(\boldsymbol{\theta})$ that we denote $Y(\boldsymbol{\theta}) = \mathbf{S}_{i,[b]}\hat{Y}(\boldsymbol{\theta})_{+}$.

For a differentiable learning objective $\mathcal{L} : \mathbb{R}^{|\boldsymbol{\theta}|} \mapsto \mathbb{R}$, the loss gradient with respect to the network parameters $\boldsymbol{\theta}$ can be obtained as

$$\begin{aligned}
\nabla_{\boldsymbol{\theta}}\mathbb{E}_Y\left[\mathcal{L}\left(Y(\boldsymbol{\theta})\right)\right] &= \nabla_{\boldsymbol{\theta}}\left[\int_Y \mathcal{L}\left(Y(\boldsymbol{\theta})\right)\mathbb{P}(Y|\boldsymbol{\theta})\delta Y\right] \\
&= \int_Y \nabla_{\boldsymbol{\theta}}\mathcal{L}\left(Y(\boldsymbol{\theta})\right)\mathbb{P}(Y|\boldsymbol{\theta})\delta Y + \int_Y \mathcal{L}\left(Y(\boldsymbol{\theta})\right)\nabla_{\boldsymbol{\theta}}\mathbb{P}(Y|\boldsymbol{\theta})\delta Y \\
&= \mathbb{E}_Y\left[\nabla_{\boldsymbol{\theta}}\mathcal{L}\left(Y(\boldsymbol{\theta})\right)\right] + \int_Y \mathcal{L}\left(Y(\boldsymbol{\theta})\right)\nabla_{\boldsymbol{\theta}}\mathbb{P}(Y|\boldsymbol{\theta})\delta Y.
\end{aligned} \tag{29}$$

Note that the expectation's gradient is not the expectation of the gradient due to the second term that captures the dependence of the probability of $Y(\boldsymbol{\theta})$ on the parameters of the network. The second term may face challenges when the probability of $Y(\boldsymbol{\theta})$ does not have an analytical form or its gradient $\nabla_{\boldsymbol{\theta}}\mathbb{P}(Y|\boldsymbol{\theta})$ poses challenges for automatic differentiation tools. The solution proposed by Kingma & Welling (2013) reparameterizes the target variable to recover the differentiability of the gradient loss in Eqn. 29.

Using the chain rule and the reparameterization trick, the loss' gradient can be recovered:

$$\begin{aligned}
\nabla_{\boldsymbol{\theta}}\mathbb{E}_Y\left[\mathcal{L}\left(Y(\boldsymbol{\theta})\right)\right] &= \nabla_{\boldsymbol{\theta}}\mathbb{E}_{\boldsymbol{\epsilon},\mathbf{z}}\left[\mathcal{L}\left(\mathbf{S}_{[i][b]}\hat{Y}(\boldsymbol{\theta})\right)\right] \\
&= \mathbb{E}_{\mathbf{z},\boldsymbol{\epsilon}}\left[\nabla_{\boldsymbol{\theta}}\mathcal{L}\left(\mathbf{S}_{i,[b]}\hat{Y}(\boldsymbol{\theta})\right)\right] \\
&= \mathbb{E}_{\mathbf{z},\boldsymbol{\epsilon}}\left[\frac{\delta\mathcal{L}}{\delta Y} \cdot \mathbf{S}_{i,[b]} \cdot \left(\nabla_{\boldsymbol{\theta}}\hat{Y}(\boldsymbol{\theta})\right)\right] \\
&= \mathbb{E}_{\mathbf{z},\boldsymbol{\epsilon}}\left[\frac{\delta\mathcal{L}}{\delta Y} \cdot \mathbf{S}_{i,[b]} \cdot \left[\nabla_{\boldsymbol{\theta}}\mu + \nabla_{\boldsymbol{\theta}}\ \sigma\ \mathbf{z} + \nabla_{\boldsymbol{\theta}}\mathbf{F}\boldsymbol{\epsilon}\right]_{+}\right].
\end{aligned} \tag{30}$$

As highlighted in Section 5, the key insight from Eqn. 30 is that the reparameterization trick provides a highly flexible framework. It is compatible with any differentiable learning objective and can accommodate differentiable constraints that extend beyond traditional aggregation constraints, such as those in Eqn. 28. Furthermore, the techniques explored in this paper can be generalized to continuous distributions (Figurnov et al., 2018; Ruiz et al., 2016; Jankowiak & Obermeyer, 2018) beyond Gaussian random variables, opening up exciting avenues for future research.

