# OpenReview forum: "$\clubsuit$ CLOVER $\clubsuit$: Probabilistic Forecasting with Coherent Learning Objective Reparameterization"
_TMLR — Accepted by TMLR_

### Review · Reviewer_jJ2e · 2024-09-15

**Summary Of Contributions:**

The paper presents a probabilistic model for hierarchical time series prediction. The model uses a parametrization matrix to predict all (linear) aggregations from the base predictions. Additionally, a new network structure based on temporal convolutions is introduced and the classical, but not well-known loss function CRPS is being used. Experiments are conducted that show the advantage of this approach.

**Audience:**

Yes

**Claims And Evidence:**

Yes

**Requested Changes:**

You should clearly improve your experiments!

- In general, the introduction seems excessively long with 3.5 pages. You might split everything after your contributions into a new section. Or you cold write the introduction shorter.
- 2nd contribution: aren't these multiple contributions combined in one long paragraph?
- Is there no joint probability between time steps in (3)? I.e. are your predictions conditionally independent predictions after conditioning on the past?
- While notations (p.3) is mostly clear, somewhat more context would help, like writing some more invariants like $[i] = [a]\uplus [b]$ or $|[a]|=N_a$ etc.
- What is a probabilistic forecast space? Is this a probability space (as vagely defined in Definition 1.1)? (If it is the latter case, one usually writes "(Omega,F,P)" instead of "Omega,F,P".) Since probabilistic forecast space appears again later, it should be formally be defined.
- Definition 1.1: What is a "set’s image"? Under which map? The projection on the relevant coordinates?
- Please introduce CRPS and clearly motivate its main idea and advantage.
- Various tables: how many standard deviations are you giving in your error estimations?
- Table 3: regarding the missing results: can you extend the implementation, that prevented these results? Or can you produce the missing results for at least on hierarchy?
- I do not understand Figure 3. Which data sets are being used? How many repetitions?
- In Section 3.3, please directly reference the tables and figures with the corresponding results and include these results in the main paper. (You only reference to APpendix F for details, not results.) I find this very confusing.

**Strengths And Weaknesses:**

Strengths
- The (long) introduction  is well-written
- Methods are clearly decribed, down to hyperparameters and including some well-formated code.

Weaknesses
- The rest of the paper, apart from the introduction but including the abstract, could use some polish in writing style. While is did not find too many errors, it was hard for me to follow sometimes. Additional points see below.
- The experiments do not show everything that is being claimed:
  - 3rd contribution: while you show better accuracy than the state of the art, you do not demonstrate "the adaptability to fit complex temporal relationships between known data and future predictions, while also modeling correlations between elements in the hierarchy" later in the paper
  - I think Table 3 and Figure 3.a (and more tables and figures later) are misleading. You validate on CRPS. Of course, this validation works best, if you train on CRPS and not on different criteria. (I know that e.g. Table 4 shows additional results that do not have this problem.)
  - I do not understand how much of the gain by this method comes from the loss function and how much from certain parts of the model architecture. While you make some ablations, that part could be clearer (and be described more clearly). Why using Convolutions? Why
- Just a combination of known techniques, less fancy (but ok for TMLR)

---

> ### Author Response · Authors · 2024-11-05
> **Strengths**
>
> **The (long) introduction is well-written, the methods are clearly described, down to hyperparameters and including some well-formatted code.**
>
> We appreciate that the reviewer recognizes the merits of the paper and found it well-written. We have now addressed the remaining issues (details below). For convenience, we have marked all changes in the manuscript in **RED**.
> See anonymized paper: https://openreview.net/forum?id=SE2nigS2ad

---

> ### Author Response · Authors · 2024-11-05
> **Weaknesses**
>
> **The rest of the paper, apart from the introduction but including the abstract, could use some polish in writing style. While it did not find too many errors, it was hard for me to follow sometimes. Additional points see below. The experiments do not show everything that is being claimed.**
>
> - **A. 3rd contribution: while you show better accuracy than the state of the art, you do not demonstrate "the adaptability to fit complex temporal relationships between known data and future predictions, while also modeling correlations between elements in the hierarchy" later in the paper.**
>
>     We improved contribution 2) and made more precise the claims in the introduction.
>
>     Additionally, to support the claim of the capabilities of our model to estimate correlations between the elements in the hierarchy  we added Figure 8 to Appendix E.  Figure 8 shows the estimated correlation matrices produced by our model.
>
> - **B. I think Table 3 and Figure 3.a (and more tables and figures later) are misleading. You validate on CRPS. Of course, this validation works best, if you train on CRPS and not on different criteria. (I know that e.g. Table 4 shows additional results that do not have this problem.).**
>
>     Thank you for your feedback. We apologize for any confusion and would like to clarify that aligning the CRPS training and evaluation is not the only contribution of our paper. To improve clarity of our work and contributions:
>
>     - B.1 We improved the captions in Table 3 and Figure 3.a (now Figure 3.b).
>
>     - B.2 We refined the description of our contributions, now clearly stating: 1) the development of a coherent factor model applicable to most neural forecasting architectures, 2) the introduction of a CRPS learning objective for better train/evaluation alignment, 3) VAR-input extensions to the MQCNN architecture, and 4) achieving state-of-the-art results in hierarchical forecasting.
>
>     - B.3 In Appendix F, we expanded the ablation study of network architectures, to clearly identify the source of improvements in our method, see response C below.
>
>     - B.4 In Appendix G, Table 7 (previous Table 4), we also present mean forecast evaluations, showing that the CRPS learning objective not only improves CRPS performance but also enhances mean forecasts.
>
>     - B.5 Additionally, as we now state in Section 2, CRPS has been the standard evaluation metric in hierarchical forecasting for the past decade, and its use in our work is consistent with established literature.
>
> - **C. I do not understand how much of the gain by this method comes from the loss function and how much from certain parts of the model architecture. While you make some ablations, that part could be clearer (and be described more clearly). Why use Convolutions? Why?**
>
>     To further clarify the source of the observed performance improvements we updated the ablation study in Appendix F, Table5. We now compare well-performing neural forecasting architectures, TFT, NBEATS, NHITS, and LSTM (univariate)  and FC-GAGA (multivariate), all augmented with the Factor Model. This experiment confirms that convolution-based neural architectures continue to deliver state-of-the-art results. As shown in Figure 3 MQCNN is on par with the univariate architectures and MQCNN+CrossSeriesMLP significantly improves upon them.
>
>     Table 5 ablation shows that a significant portion of the performance gain can be attributed to the inclusion of the CrossSeriesMLP, which brings VAR-like capabilities to the MQCNN, further enhancing its predictive power across series.

---

> > ### Author Response · Authors · 2024-11-05
> > **Requested Changes**
> >
> > **You should clearly improve your experiments!**
> >
> > We appreciate the feedback, and improved the experiments in the revision.
> > We doubled the amount of datasets of the main experiments, and significantly improved the ablation studies in Appendix F to identify clearly the sources of improvement in our methodology.
> >
> > - **A. In general, the introduction seems excessively long with 3.5 pages. You might split everything after your contributions into a new section. Or you could write the introduction shorter.**
> >
> >     Thanks for the feedback. We reduced Section 1 introduction and moved part of the introduction into a new Section 2 dedicated to the formulation of the hierarchical forecast task. This way we reduced almost by half Section 1's introduction.
> >
> > - **B. 2nd contribution: aren't these multiple contributions combined in one long paragraph?**
> >
> >     We thank the reviewer for the valuable suggestion, we’ve reorganized the contribution section, and splitted these contributions to 1) multivariate factor model that can capture complex correlation structure, 2) joint probabilistic model that optimizes for CRPS evaluation metric, 3) VAR-augmented MQCNN architecture, and 4) achieving state-of-the-art accuracy results on three publicly available datasets.
> >
> > - **C. Is there no joint probability between time steps in (3)? I.e. Are your predictions conditionally independent predictions after conditioning on the past?**
> >
> >     That is correct, we don't model the joint distribution considering relationships of the forecast across the forecast creation date. The conditional independence assumption across time in Eqn (3), maintains the computational tractability of the forecasting task, and is an assumption used by most forecasting methods. We added a footnote to Eqn (3) explaining this.
> >
> > - **D. While notations (p.3) is mostly clear, somewhat more context would help, like writing some more invariants like [i]=[a]+[b]?**
> >
> >     We incorporated the suggestion and complemented the definition of the hierarchical notation indexes in Section 2.
> >
> > - **E. What is a probabilistic forecast space? Is this a probability space (as vaguely defined in Definition 1.1)? (If it is the latter case, one usually writes "(Omega,F,P)" instead of "Omega,F,P".) Since probabilistic forecast space appears again later, should it be formally defined?**
> >
> >     Thanks for the suggestions, we improved Definition 2.1 on the coherent probabilistic forecast space. By a) using parentheses on the triplets defining the space; b) linking the definition of coherence with Eqn (3) that states the classic forecast problem. The definition of the forecast space is implied by Eqn (3).
> >
> > - **F. Definition 1.1: What is a "set’s image"? Under which map? The projection on the relevant coordinates?**
> >
> >     We now describe the linear transformation used in the definition of the coherent forecast space. In particular Definition 2.1’s relationship to the aggregation constraints matrix.
> >
> > - **G. Please introduce CRPS and clearly motivate its main idea and advantage.**
> >
> >     We thank the reviewer for the suggestion. We have added a new paragraph in Section 2 to introduce this commonly used evaluation metric and its advantages.
> >
> > - **H. Various tables: how many standard deviations are you giving in your error estimations?**
> >
> >     We improve the description of Table 3, 5, 6 and 7, to include the definition of the 95% confidence interval (1.96 times standard deviation).
> >
> > - **I. Table 3: regarding the missing results: can you extend the implementation that prevented these results? Or can you produce the missing results for at least one hierarchy?**
> >
> >     Extending the PERMBU for the TourismL case is a contribution beyond the scope of the paper, and would merit its own dedicated publication. We improve Table 3’s missing explanation that now reads: “PERMBU-MinT on TourismL is unavailable because its implementation cannot naively be applied to datasets with multiple hierarchies, since PERMBU’s cannot identify a unique way to obtain the upper level distributions. ”
> >
> > - **J. I do not understand Figure 3. Which data set is being used? How many repetitions?**
> >
> >     We improved the description of the ablation studies reported in Figure 3, mentioning the Traffic dataset and the amount of times we measure the validation signal.
> >
> > - **K. In Section 3.3, please directly reference the tables and figures with the corresponding results and include these results in the main paper. (You only reference Appendix F for details, not results.) I find this very confusing.**
> >
> >     We apologize for the confusion. We have revised Section 4.3 to provide a concise summary of the ablation study results, detailed in Appendix F. To improve clarity, we now directly reference Tables 5 and 6 from Appendix F within the main text, ensuring the key results are highlighted without requiring readers to navigate to Appendix F for essential information.

---

> ### Comment · Reviewer_jJ2e · 2024-11-07
> **Answer to the rebuttal**
>
> My concerns are adequately adressed. To my (superficial) opinion, the questions by the other reviews are also adequately adressed.
>
> I think the paper is ready to being published.

---

### Review · Reviewer_3wgV · 2024-10-15

**Summary Of Contributions:**

This work proposes a new method for hierarchical time series forecasting. The paper makes the following contributions:
- Proposal of Deep Coherent Factor Model Neural Network (DeepCoFactor) model, a method for probabilistic coherent forecasting. DeepCoFactor can directly optimize metrics of interest such as CRPS, MSE, or others rather than naively maximizing the log likelihood which might not always provide the best generalization under model misspecification. This makes the proposed approach versatile.
- The paper has basic grounding in theory (Lemma A.2) and offers strong experimental results on 3 datasets.

**Audience:**

Yes

**Claims And Evidence:**

Yes

**Requested Changes:**

## Dataset Choice
Can the authors justify the dataset choice or provide experiments with more datasets? Even a controlled toy experiment might prove useful to explain where the performance improvement of their method comes from and what generalizable insights we can draw about model performance beyond the selected datasets. Some dataset ideas come from the baselines, e.g., in HIER-E2E (see paper for full bibliographical citation): Labour, Tourism, Wiki.

## Baseline selection
Can the authors explain why they do not compare to "HiPerformer" [https://arxiv.org/pdf/2305.08073]? Is a comparison not applicable? If it is can they revise the paper to compare to this approach?

## Additional Metrics:
As mentioned in weaknesses section, I have my concerns about reporting rSE. In general, squared error based metrics can overestimate the importance of outliers. Just because a method does better wrt to outliers does not mean that the method is better overall. A very useful metric that is less prone to noise from outliers is symmetric mean absolute percent error sMAPE. This metric would 100% convince me that the proposed method is indeed better, at least for point predictions. For distributional predictions, CRPS is a good metric and I am convinced that the proposed method is strong here. An nice-to-have metric to check would be 95% C.I. coverage (how often is the predicted interval between the 2.5th and 97.5th percentile covering the ground truth)? How does this compare to baselines?

## Notation
Notation is generally a strong point of the paper. However I have one question.
- Definition 1.1: "[...] for any set \mathcal{B} \in \mathbb{F}_{[b]}". What is \mathbb{F}_{[b]}? My understanding is that \mathbb{P} maps elements of the sigma algebra to real valued scalar / vector. Should the notation be \mathcal{B} \in \mathcal{F}_{[b]} instead?

## Rephrase / Typos:
- Some phrases that need editing: "DeepCoFactor leverages coherently aggregates the samples of [...]".
- Some typos, e.g., in first bullet of contributions, last sentence: "[...] specializes in mult-step probabilistic forecast".
Please edit the text to fix these.

## Additional questions:
- In Table 1 the authors claim that baseline method HIER-E2E (see paper for full bibliographical citation) is does not model a joint multivariate probability distribution. However, HIER-E2E claims to be a multivariate method. Is something unclear to me or do these two statements contradict?
- The paper borrows a lot of definitions and examples at the end of the introduction from DPMN baseline (see paper for full bibliographical citation). Figures 1 and 2 are very similar to DPMN as well. While I do not consider this plagiarism, if definitions and the example of Fig. 1 are directly taken from DPMN it should be made clear that the definitions are being quoted from that paper with proper attribution.
For example, "We can formalize this through Definition 1.1, which essentially tells us that the distribution of a given aggregate random variable is exactly the distribution defined as [...]". I would rephrase this sentence to point to the fact that definitions are taken from the literature as it might mislead the audience into thinking that this definition is original. Additionally, to make the paper more novel I would generate I completely new example in figure 1 as it is basically identical to DPMN example. Furthermore, I would explain that Fig. 2 mirrors style of DPMN architecture figure to highlight differences between the model inputs, internals and outputs. Overall the similarity in Fig. 2 makes the novelty of this paper seem smaller than it actually is.

**Strengths And Weaknesses:**

My general opinion of the paper is positive, I think that the topic is interesting and the claims are likely true given the empirical results. However, there are some concerns mostly with writing and experimental design. If enough of my concerns are addressed I would lean towards accepting the paper.

Strengths:
- Interesting problem, relevant to TMLR community.
- The proposed architecture does what it promises, it models hierarchical time series and aggregates them coherently by design.
- Strong experimental results. The metrics often improve above 5%.
- Notation and writing are clear, though there are occasional typos (see Rephrase / Typos section in "Requested Changes" part of this review). Note this statement is qualified by the fact that I have a single strong objection to the writing in the introduction section. Please see Additional questions under "Requested Changes" part of the review. All other parts of the paper are clear and well written.

Weaknesses:
- Limited datasets: It seems that the datasets that the authors compare on are only a subset of the datasets explored from the baselines. Why is this?
- Selection of forecast metrics: CRPS is an excellent metric to report. However, I have my concerns about the usefulness of reporting rSE. In general, squared error based metrics can overestimate the importance of outliers. Just because a method does better wrt to outliers does not mean that the method is better in general. Please see requested change for additional metrics.
- Baseline selection: It seems that a new model "HiPerformer" [https://arxiv.org/pdf/2305.08073] compares on similar datasets and baselines.

---

> ### Author Response · Authors · 2024-11-05
> **Strengths**
>
> **My general opinion of the paper is positive, I think that the topic is interesting and the claims are likely true given the empirical results. However, there are some concerns mostly with writing and experimental design. If enough of my concerns are addressed I would lean towards accepting the paper.**
>
> - **A. Interesting problem, relevant to the TMLR community.**
> - **B. The proposed architecture does what it promises, it models hierarchical time series and aggregates them coherently by design.**
> - **C. Notation and writing are clear, though there are occasional typos (see Rephrase / Typos section in "Requested Changes" part of this review). Note this statement is qualified by the fact that I have a single strong objection to the writing in the introduction section. Please see Additional questions under "Requested Changes" part of the review. All other parts of the paper are clear and well written.**
>
> We appreciate that the reviewer recognizes the paper's merits and finds it interesting and relevant for the TMLR community. We have addressed the requested changes, and marked the changes in **RED**. See anonymized paper: https://openreview.net/forum?id=SE2nigS2ad

---

> > ### Author Response · Authors · 2024-11-05
> > **Weaknesses**
> >
> > - **A. Limited datasets: It seems that the datasets that the authors compare on are only a subset of the datasets explored from the baselines. Why is this?**
> > - **B. Selection of forecast metrics: CRPS is an excellent metric to report. However, I have my concerns about the usefulness of reporting rSE. In general, squared error based metrics can overestimate the importance of outliers. Just because a method does better wrt to outliers does not mean that the method is better in general. Please see requested change for additional metrics.**
> > - **C. Baseline selection: It seems that a new model "HiPerformer" [https://arxiv.org/pdf/2305.08073] compares on similar datasets and baselines.**
> >
> > Thank you for your feedback. We have addressed all the points raised in the reviews, including dataset selection, forecast metrics, updated literature, and baseline comparisons (detailed below). We hope the revised version meets your expectations.

---

> > > ### Author Response · Authors · 2024-11-05
> > > **Dataset Choice**
> > >
> > > **Can the authors justify the dataset choice or provide experiments with more datasets? Even a controlled toy experiment might prove useful to explain where the performance improvement of their method comes from and what generalizable insights we can draw about model performance beyond the selected datasets. Some dataset ideas come from the baselines, e.g., in HIER-E2E (see paper for full bibliographic citation): Labour, Tourism, Wiki.**
> > >
> > > We appreciate the suggestion, we extended the main results in Table 3 and relSE results in AppendixF, Table 7 to include the Labour, TourismSmall, and Wikipedia datasets from HierE2E (Rangapuram, et al., 2021) The paper modifies several sections including the introduction, the conclusion, Appendix D on dataset details and Appendix C on hyperparameters to account for the inclusion of the suggested datasets.

---

> ### Author Response · Authors · 2024-11-05
> **Baseline Selection**
>
> **Baseline selection Can the authors explain why they do not compare to "HiPerformer" [https://arxiv.org/pdf/2305.08073]? Is a comparison not applicable? If it is, can they revise the paper to compare to this approach?**
>
> We appreciate the reviewer’s suggestion, and added the Hiperformer (Umagami, et al. 2023) to our literature review. Hiperformer’s predicted distributions following Equation (2) are not strictly coherent. As we mention in Section1, paragraph 4, in this paper we focus our evaluation to strictly coherent methods. Future research can include a comparison between coherent and soft coherent methods.
>
> In addition to the coment above on the Hiperformer architecture, we significantly improved the ablation studies in Appendix F. Now we compare our model with MQCNN, TFT, NBEATS, NHITS, LSTM and a task specialized FCGAGA architecture, all augmented with the coherent factor model distribution.

---

> ### Author Response · Authors · 2024-11-05
> **Additional Metrics**
>
> **As mentioned in the weaknesses section, I have my concerns about reporting rSE. In general, squared error based metrics can overestimate the importance of outliers. Just because a method does better wrt to outliers does not mean that the method is better overall. A very useful metric that is less prone to noise from outliers is symmetric mean absolute percent error sMAPE. This metric would 100% convince me that the proposed method is indeed better, at least for point predictions. For distributional predictions, CRPS is a good metric and I am convinced that the proposed method is strong here. A nice-to-have metric to check would be 95% C.I. coverage (how often is the predicted interval between the 2.5th and 97.5th percentile covering the ground truth)? How does this compare to baselines?**
>
> - A. We appreciate the suggestion. To evaluate the forecast calibration/coverage we added a probability-probability plot in Figure 7. Figure 7 complements Appendix F on the “Forecast Distributions Visualization”. The qualitative evaluation of the calibration/coverage in Figure 7, shows a very reasonable similarity of our model’s forecast distribution and the empirical distribution.
>
> - B. While we appreciate the sMAPE suggestion, we don’t want point forecasting to be distracting from the main results of the paper, which focuses on probabilistic forecasting. We moved relSE to the Appendix G.
> In addition sCRPS is closely correlated to sMAPE due to their relationship to QL, and their results would be redundant. We added a phrase with this explanation and our motivation for relSE in Appendix G.
>
> - C. We acknowledge that squared error metrics can be sensitive to outliers; however, relSE allows us to compare with much of the hierarchical forecasting literature, which is largely dominated by mean reconciliation strategies such as Bottom-Up, Top-Down, MinTrace, and ERM and report mostly relSE.

---

> > ### Author Response · Authors · 2024-11-05
> > **Notation**
> >
> > **Notation is generally a strong point of the paper. However I have one question. Definition 1.1: "[...] for any set \mathcal{B} \in \mathbb{F}{[b]}". What is \mathbb{F}{[b]}? My understanding is that \mathbb{P} maps elements of the sigma algebra to real valued scalar / vector. Should the notation be \mathcal{B} \in \mathcal{F}_{[b]} instead?**
> >
> > We appreciate such a careful revision. We fixed Equation 4’s typo from \mathbb{F} to \mathcal{F} that denotes our forecast distributions event space.

---

> > > ### Author Response · Authors · 2024-11-05
> > > **Rephrase / Typos:**
> > >
> > > - **A. Some phrases that need editing: "DeepCoFactor leverages coherently aggregates the samples of [...]".**
> > >
> > >     We appreciate such a careful revision. We fixed the typo on Figure 2’s caption.
> > >
> > > - **B. Some typos, e.g., in first bullet of contributions, last sentence: "[...] specializes in multi-step probabilistic forecast". Please edit the text to fix these.**
> > >
> > >     We revised the contributions summary to clearly highlight: 1) the multivariate factor distribution, 2) CRPS optimization for the factor model, 3) the VAR-augmented MQCNN architecture, and 4) state-of-the-art results. The new point 3, our updated description of the DeepCoFactor architecture no longer references MQForecaster’s specialization in multi-step forecasts.

---

> > > > ### Author Response · Authors · 2024-11-05
> > > > **Additional questions**
> > > >
> > > > - **A. In Table 1 the authors claim that baseline method HIER-E2E, it does not model a joint multivariate probability distribution. However, HIER-E2E claims to be a multivariate method. Is something unclear to me or do these two statements contradict?**
> > > >
> > > >     HIER-E2E (https://proceedings.mlr.press/v139/rangapuram21a/rangapuram21a.pdf) receives VAR-like inputs but does not model multivariate correlations/outputs. We reference HierE2E’s Section 3.1 of the paper, which states: "The bottom-level time series are too sparse to learn any covariance structure, let alone more complicated nonlinear relationships between them. Given this, we propose to learn a diagonal covariance matrix Σt …"
> > > >
> > > >     We modified Table 1 to distinguish between the multivariate inputs and multivariate output properties.
> > > >
> > > > - **B. The paper borrows a lot of definitions and examples at the end of the introduction from DPMN baseline (see paper for full bibliographical citation). Figures 1 and 2 are very similar to DPMN as well. While I do not consider this plagiarism, if definitions and the example of Fig. 1 are directly taken from DPMN it should be made clear that the definitions are being quoted from that paper with proper attribution. For example, "We can formalize this through Definition 1.1, which essentially tells us that the distribution of a given aggregate random variable is exactly the distribution defined as [...]". I would rephrase this sentence to point to the fact that definitions are taken from the literature as it might mislead the audience into thinking that this definition is original. Additionally, to make the paper more novel I would generate I completely new example in figure 1 as it is basically identical to DPMN example. Furthermore, I would explain that Fig. 2 mirrors style of DPMN architecture figure to highlight differences between the model inputs, internals and outputs. Overall the similarity in Fig. 2 makes the novelty of this paper seem smaller than it is**
> > > >
> > > >     - Thank you for your comment. We have added additional references to Olivares et al. (2023). Section 2 now includes a header: "In this section, we introduce the hierarchical forecasting task following the work of \citet{Olivares2021ProbabilisticHF}." In addition we have two paragraphs on Section 3.4 discussing the key differences of this work and DPMN.
> > > >     - Thank you for your suggestion to highlight modifications to the DPMN architecture and mark our contributions in Figure 2 in red to make these distinctions clearer. We understand your concerns, but we want to clarify that the similarity in Figures 2 and and DPMN’s Figure 3 is intentional, as it helps to emphasize the key contributions of our work:  the CrossSeriesMLP enabling VAR-inputs to the MQCNN architecture and the reparameterization trick, which enable alternative learning objectives. We believe this approach highlights the differences more effectively.

---

> ### Comment · Reviewer_3wgV · 2024-11-07
> **Some additional comments**
>
> I would like to thank the authors for the thorough response to my comments. The paper is much improved after the edits, especially with the additional baseline methods, datasets and ablations.
>
> A couple of final comments to address:
>
> 1. In the newest revision in Definition 2.1 "[..] with sample space $(\Omega_{[b]}, \mathcal{F}_{[b]})$ and [...]":
>
> The standard definition of a probability space would denote the sample space as $\Omega_{[b]}$ (not the tuple above). $\mathcal{F}_{[b]}$ would be the event space. The event space is a set of events with each event being a set of outcomes in the sample space.
> Please update the definition to reflect that.
>
> 2. Do the authors commit to making their code and experiments public? I think *open access to code is very important* for the TMLR community. This is my last concern for the paper, before I can make my recommendation.

---

> > ### Author Response · Authors · 2024-11-07
> >
> > - **The standard definition of a probability space would denote the sample space as (not the tuple above). Wwould be the event space. The event space is a set of events with each event being a set of outcomes in the sample space. Please update th definition to reflect that.**
> >
> >     Thanks again, for such a detailed review. We just fixed the definition.
> >
> >
> > - **Do the authors commit to making their code and experiments public? I think open access to code is very important for the TMLR community. This is my last concern for the paper, before I can make my recommendation.**
> >
> >     Thank you for the recommendation. In the camera-ready version, we will provide links to the preprocessing, wrangling of public benchmark datasets, baseline methods, and forecast pipelines to facilitate other researchers interested in the topic.
> >
> >     We understand the importance of open access and are doing our best to make as much of the project available as possible.
> >     However, we cannot commit to share proprietary code as it would be considered illegal at this stage. *The TMLR publication would certainly support the code sharing internal processes.*
> >
> >     That being said, it is our understanding that this is not a requirement of the TMLR journal for publication. Please let us if we are wrong about this.

---

> > > ### Comment · Reviewer_3wgV · 2024-11-07
> > >
> > > I appreciate that the authors agreed to do their best to make as much of the project code available as possible in the final version of the paper (preprocessing, wrangling of public benchmark datasets, baseline methods, and forecast pipelines to facilitate other researchers interested in the topic).
> > >
> > > Based on the response from the authors, my understanding is that at least some of them work in industry in which case part of the code is proprietary and cannot be shared publicly so I will not insist further on this point.
> > >
> > > It also seems that the points raised by other reviewers have been addressed.
> > >
> > > I am in favor of publishing the paper.

---

### Review · Reviewer_RBzB · 2024-10-22

**Summary Of Contributions:**

The focus of the paper is to perform hieratical probabilistic forecasting of time series. To take advantage of the hierarchy between the forecasted quantities, the proposed approach takes a MQForecaster neural network architecture and augments it with a  deep Gaussian factor forecasting model. The proposed approach achieves promising performance when compared to the state of the art.

**Audience:**

Yes

**Broader Impact Concerns:**

None.

**Claims And Evidence:**

Yes

**Requested Changes:**

See weaknesses above.

**Strengths And Weaknesses:**

Strengths:
* The proposed method is novel and interesting.

* The proposed approach shows promising performance.

* The paper is generally well written.

Weakness:
* The paper only considers aggregate time-series which are linear combinations of the base time-series (Eq. 1). Can the proposed approach deal with time-series which are composed on non-linear aggregates of the base time-series?

* The paper says that the HierE2E method of Rangapuram et al. (2021) is “too general". Could the HierE2E model be better regularized with more data? How does the proposed approach compare to HierE2E  with increasing amounts of available training data?

* Real-world datasets: The paper only considers small scale datasets such as Traffic. It would be interesting to see if the proposed approach scales to larger datasets such as nuScenes.

* The paper could consider simpler transformer based approaches as baselines.

* Few typos: Eq. 10.

Overall:
* A great paper with a novel approach and good results.

---

> ### Author Response · Authors · 2024-11-05
> **Strengths**
>
> **The focus of the paper is to perform hieratical probabilistic forecasting of time series. To take advantage of the hierarchy between the forecasted quantities, the proposed approach takes a MQForecaster neural network architecture and augments it with a deep Gaussian factor forecasting model. The proposed approach achieves promising performance when compared to the state of the art. Overall: A great paper with a novel approach and good results.**
> - **The proposed method is novel and interesting.**
> - **The proposed approach shows promising performance.**
> - **The paper is generally well written.**
>
> We appreciate the positive evaluation. We have incorporated the suggestions in the revised paper and marked the changes in the paper in **RED**.
> See anonymized paper: https://openreview.net/forum?id=SE2nigS2ad

---

> ### Author Response · Authors · 2024-11-05
> **Weaknesses**
>
> * **A. The paper only considers aggregate time-series which are linear combinations of the base time-series (Eq. 1). Can the proposed approach deal with time-series which are composed on non-linear aggregates of the base time-series?**
>
>     Thank you for the insightful question. While handling non-linear aggregates is beyond the current scope of the paper, we have added this point to the discussion section:
>     "Another direction for future research is to extend the usage of the reparameterization trick from the learning objectives, into the hierarchical aggregation structure itself, provided the aggregation structure is done through differentiable transformations."
>
>
> * **B. The paper says that the HierE2E method of Rangapuram et al. (2021) is “too general". Could the HierE2E model be better regularized with more data? How does the proposed approach compare to HierE2E with increasing amounts of available training data?**
>
>     We already compare DeepCoFactor with the HierE2E model on the large-scale Favorita experiment. The HierE2E model presents significant computational challenges in large-scale comparisons:
>
>     * HierE2E is implemented in MXNet, supports only CPU, and has not been updated in three years, making large-scale experiments time-consuming. (https://github.com/rshyamsundar/gluonts-hierarchical-ICML-2021)
>     * Its LSTM-based encoder and decoder result in particularly slow inference compared to more efficient multi-step forecast strategy from DeepCoFactor.
>
> * **C. Real-world datasets: The paper only considers small scale datasets such as Traffic. It would be interesting to see if the proposed approach scales to larger datasets such as nuScenes.**
>
>     We want to highlight that our main experiment features the large-scale Favorita grocery sales dataset, which includes 4036 items across 93 locations, totaling 375,348 series. Additionally, in this revision, we've expanded our experiments from three to six datasets, including the recently added TourismS, Traffic, and Wikipedia datasets. As far as we can tell, nuScenes is not a dataset easily amenable to forecasting, and would require significant effort to bring into our problem setting.
>
> * **D. The paper could consider simpler transformer based approaches as baselines.**
>
>     Thank you for the suggestion. In our ablation study (Appendix F, Table 5), we compare several strong neural forecasting architectures, including TFT, N-BEATS, N-HiTS, LSTM (univariate), and FC-GAGA (multivariate), all augmented with the Factor Model. TFT (Lim et al., 2021, https://arxiv.org/abs/1912.09363) is a well-established transformer-based method. The study shows MQCNN performs on par with or better than most univariate architectures, with MQCNN+CrossSeriesMLP significantly outperforming them.

---

> > ### Comment · Reviewer_RBzB · 2024-11-25
> > **Good Rebuttal**
> >
> > The rebuttal addresses most of my concerns.

---

### Author Response · Authors · 2024-11-05
**Revision Summary**

We are pleased that the Action Editor and the Reviewers appreciate our manuscript. We have addressed all points raised in the reviews. We hope that you find the current version ready to be published.

In response to the feedback we have taken the opportunity to improve the paper’s main experiments, and flow. The key changes in the updated manuscript are:
- We improved the contributions, and introduction, reducing it from the previous four pages to one. We added Section 2, dedicated to the hierarchical forecast notation.
- We doubled the number of datasets in the main experiments in Section 4. Now the main experiments include Australian Labour, monthly Tourism, quarterly Tourism, Favorita sales, SF Traffic and Wikipedia article visits datasets.
- We extended the ablation studies, to compare our model to well-established neural architecture alternatives, including TFT, NBEATS, NHITS, LSTM, and FC-GAGA.  We extended ablation study and main experiments to include results from our model trained with the energy score.

For convenience we highlighted the paper changes in **RED**.
See anonymized paper: https://openreview.net/forum?id=SE2nigS2ad

Sincerely yours,
The Authors

---

### Decision · Action_Editor_iUp6 · 2024-11-29

**Recommendation:** Accept with minor revision

**Comment:**

The paper proposes a novel approach that is based on interesting ideas. The methodology experiments are sufficient to support the claims.

The only minor change that is requested is a moderation of the claim in the introduction that the approach can be “based on any generic deep learning univariate forecasting model”. This should at least be moderated by the condition that the forecasting model can output factor model parameters. But in addition to this, the experiments are conducted with only a single forecasting model, so there is not really empirical evidence that the method can successfully employ any univariate forecasting model.  While “successfully” is not claimed, it is implied by the current text.

In making their recommendations, all reviewers acknowledged that the method is novel and stated that it would be of interest to the research community. One reviewer commented that further experiments on additional datasets would further solidify the work.

**Audience:**

The paper addresses an important topic that is the focus of considerable research interest. The paper presents a novel approach that will be of interest to researchers working in the field and to a subset of researchers in the broader community.

**Claims And Evidence:**

The paper introduces a method for probabilistic forecasting that focuses on settings where there is a hierarchy among the modeled entities. The main claims of the paper are that the introduced model (i) is end-to-end coherent, in the sense that it adheres to the natural hierarchy; (ii) models the joint multivariate probability distribution; and (iii) generates differentiable samples. These main claims are supported primarily by the methodology, which is explicitly designed to achieve coherence, and is structured for multivariate modelling and differentiable samples. The paper additionally claims that the adopted approach “enhances the robustness of the probabilistic model to misspecification” and is versatile so that it can be tailored to optimize a variety of forecast metrics. The paper claims state-of-the-art performance on three public datasets. These claims are supported by experimental evidence.

The paper also claims in the introduction that the generic nature of the approach allows it to be “based on any generic deep learning univariate forecasting model”. Later in the methodology, this claim is moderated by the condition “so long as they can be made to output factor model parameters”. The paper does not convincingly demonstrate that “most univariate neural forecasting models” can be used, as is claimed in the introduction. Experiments are conducted with a limited number of models and there is not an extensive survey of univariate neural forecasting models that discusses their capabilities, especially concerning the ability to output factor model parameters. The extra condition of being able to output factor model parameters should really be included explicitly in the introduction. Otherwise, to clearly support the claim made in the introduction, the authors would need to provide evidence that most univariate neural forecasting models have this capability.

---

> ### Author Response · Authors · 2024-12-10
> **Minor Revision addressing Action Editor's feedback**
>
> We appreciate the comments. We moderated the claims in the Introduction and Section 3.2, and clarified Appendix F.
>
> Regarding the claim on "based on any generic deep learning univariate forecasting model" ... "so long as they can be made to output factor model parameters". We moderated the claim and Section 1 now reads: "This factor model is generic and can augment most neural forecasting models with minimal modifications, with the condition that all the hierarchy's series are available during training and inference."
>
> And Section 3.2 on the neural network architecture now reads: "As mentioned in Section 1, the factor model can augment most neural forecasting models if all series in the hierarchy are available during training and inference; as this is a sufficient condition to obtain the Factor model's parameters. Appendix F shows an augmented subset of Neural Forecast library models, while the main paper focuses on the MQCNN-based model.""
>
> We added this comment in Appendix F:  "We implemented the Factor Model to augment models from the Neural Forecast library, because of this it can readily augment AutoFormer, BitCN, DeepAR, DeepNPTS, DilatedRNN, DLinear, FedFormer, FCGaga, GRU, Informer, ITransformer, KAN, LSTM, MLP, NBEATS, NBEATSx, NHITS, NLinear, PatchTST, RMOK, RNN, SOFTS, TCN, TFT, Tide, TimeLLM, TimeMixer, and TimesNet. For this ablation study, we focus on NBEATS, NHITS, LSTM, FCGaga, and TFT; a comprehensive comparison of all augmented forecasting models is beyond the scope of this paper."